# GENERALIZED GUMBEL-SOFTMAX GRADIENT ESTIMATOR FOR GENERIC DISCRETE RANDOM VARIABLES

## ABSTRACT

Estimating the gradients of stochastic nodes, which enables the gradient descent optimization on neural network parameters, is one of the crucial research questions in the deep generative modeling community. When it comes to discrete distributions, Gumbel-Softmax trick reparameterizes Bernoulli and categorical random variables by continuous relaxation. However, gradient estimators of discrete distributions other than the Bernoulli and the categorical have not been explored, and the the Gumbel-Softmax trick is not directly applicable to other discrete distributions. This paper proposes a general version of the Gumbel-Softmax estimator with a theoretical basis, and the proposed estimator is able to reparameterize generic discrete distributions, broader than the Bernoulli and the categorical. In detail, we utilize the truncation of discrete random variables and the Gumbel-Softmax trick with a linear transformation for the relaxed reparameterization. The proposed approach enables the relaxed discrete random variable to be reparameterized through a large-scale stochastic computational graph. Our experiments consist of (1) synthetic data analyses and applications on VAE, which show the efficacy of our methods; and (2) topic models, which demonstrate the value of the proposed estimation in practice.

## 1 INTRODUCTION

Stochastic computational graphs, including deep generative models such as variational autoencoders, are widely used for representation learning. Optimizing the network parameters through gradient methods requires an estimation of the gradient values, but the stochasticity requires the computation of expectation, which differentiates this problem from the deterministic gradient of ordinary neural networks. There are two common ways of obtaining the gradients: score function-based methods and reparameterization methods. The score function-based estimators tend to result in unbiased gradients with high variances, while the reparameterization estimators seem to result in biased gradients with low variances (Xu et al., 2019). Hence, the core technique of the score-function based estimators becomes reducing the variances of gradients to achieve stable and fast optimizations. Meanwhile, utilizing the reparameterization estimators requires the *differentiable non-centered parameterization* (Kingma & Welling, 2014a) of random variables.

If we focus on the reparameterization estimators, one of the most popular examples is the reparameterization in the Gaussian variational autoencoder (VAE) (Kingma & Welling, 2014b), which has an exact reparameterization form. Other VAEs with explicit priors suggest the reparameterization tricks with approximations (Nalisnick & Smyth, 2017; Joo et al., 2020). For continuous random variables, it is feasible to estimate gradients with automatic differentiation by utilizing a transport equation (Jankowiak & Obermeyer, 2018) or an implicit reparameterization (Figurnov et al., 2018). However, these methods are not applicable to discrete random variables, due to the non-differentiability.

Recently, some discrete random variables, such as Bernoulli or categorical random variables, have been well-explored in terms of the reparameterization method by overcoming such difficulty through a continuous relaxation (Jang et al., 2017; Maddison et al., 2017). However, other discrete distributions have not been explored from the learning perspective in the deep generative modeling community, for example, Poisson, binomial, multinomial, geometric, negative binomial distributions, etc. Prior works on graphical models, such as Ranganath et al. (2015; 2016), utilized Poisson latent variables for the latent counting. Another line of work (Wu et al., 2020) utilized the Gaussian approximation on the Poisson latent variable to count the number of words, which can be a poor approximation

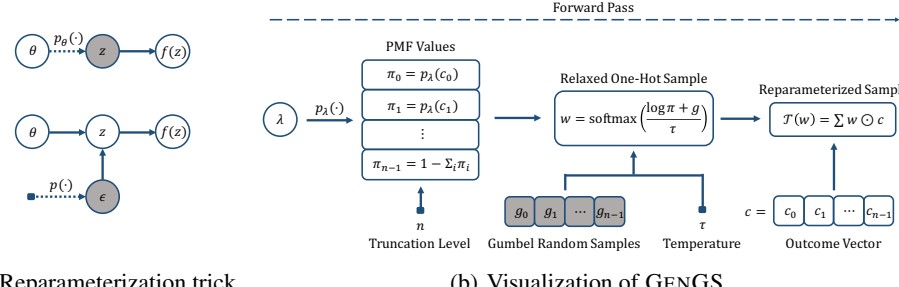

(a) Reparameterization trick.  (b) Visualization of GENGS.

Figure 1: Reparameterization in stochastic computational graphs.

if the rate parameter is small. In this sense, study on the stochastic gradient estimator for discrete distributions is required in the deep generative modeling community, which broadens the choice of prior assumptions and the utilization of various distributions.

This paper proposes a generalized version of the Gumbel-Softmax reparameterization trick, which can be utilized to generic discrete random variables through continuous relaxation, namely Generalized Gumbel-Softmax (GENGS). The key ideas of GENGS are (1) a conversion of the *sampling process* to *one-hot categorical selection process*; (2) a reversion of the *selected category in the one-hot form* to the *original sample value*; and (3) a *relaxation of the categorical selection process* into the *continuous form*. To implement these steps, GENGS first truncates discrete random variables to approximate the distribution with the finite number of possible outcomes. Afterward, GENGS utilizes the Gumbel-Softmax trick together with a special form of a linear transformation. Our main theorem supports that the proposed GENGS is applicable to *general discrete random variables, other than the Bernoulli and the categorical*. The GENGS experiments show the efficacy with synthetic examples and VAEs, as well as the usability in topic model application.

## 2 PRELIMINARY: REPARAMETERIZATION TRICK & GUMBEL-SOFTMAX

### 2.1 BACKPROPAGATION THROUGH STOCHASTIC NODES WITH REPARAMETERIZATION TRICK

Suppose we have a stochastic node, or a latent variable, $z \sim p(z|\theta)$, where the distribution depends on parameter $\theta$. The goal is optimizing the loss function $\mathcal{L}(\theta, \eta) = \mathbb{E}_{z \sim p(z|\theta)}[f_\eta(z)]$, where $f_\eta$ is a continuous and differentiable function with respect to $\eta$, i.e., neural networks. To optimize the loss function in terms of $\theta$ through the gradient methods, we need to find $\nabla_\theta \mathcal{L}(\theta, \eta) = \nabla_\theta \mathbb{E}_{z \sim p(z|\theta)}[f_\eta(z)]$, which can not be directly computed with its original form.

To compute $\nabla_\theta \mathcal{L}(\theta, \eta)$, the reparameterization trick alternatively introduces an auxiliary variable $\epsilon \sim p(\epsilon)$, which takes over all randomness of the latent variable $z$, so the sampled value $z$ can be re-written as $z = g(\theta, \epsilon)$, with a *deterministic* and *differentiable* function $g$ in terms of $\theta$. Figure 1(a) illustrates the reparameterization trick: the shaded nodes indicate random nodes, and the dotted lines denote sampling processes. Here, the gradient of the loss function with respect to $\theta$ is derived as

$$\nabla_\theta \mathcal{L} = \nabla_\theta \mathbb{E}_{z \sim p(z|\theta)}[f_\eta(z)] = \mathbb{E}_{\epsilon \sim p(\epsilon)}[\nabla_\theta f_\eta(g(\theta, \epsilon))] = \mathbb{E}_{\epsilon \sim p(\epsilon)}[\nabla_g f_\eta(g(\theta, \epsilon))\nabla_\theta g(\theta, \epsilon)] \quad (1)$$

where the last term of Equation 1 is now achievable. A condition on enabling the reparameterization trick is the assumption of the continuity of the random variable $z$, so the distribution of $z$ is limited to a class of continuous distributions. To utilize the *differentiable* reparameterization trick on discrete random variables, continuous relaxation can be applied: for example, a relaxation from the categorical distribution to the Gumbel-Softmax distribution, described in the next subsection.

### 2.2 REPARAMETERIZATION TRICK ON CATEGORICAL RANDOM VARIABLE

A *Gumbel-Max* trick (Gumbel, 1948) is a procedure for sampling a one-hot categorical value from the Gumbel distribution, instead of direct sampling from a categorical distribution. This implies that the categorical random variable $X \sim \text{Categorical}(\pi)$, where $\pi$ lies on the $(n-1)$-dimensional simplex $\Delta^{n-1}$, can be reparameterized by the Gumbel-Max trick: (1) sample $u_j \sim \text{Uniform}(0, 1)$

to generate a gumbel sample $g_j = -\log(-\log u_j)$ for each $j = 1, \cdots, n$; and (2) compute $k = \operatorname{argmax}_{j=1}^{n}[\log \pi_j + g_j]$, where $\pi$ is a categorical parameter. This procedure generates a one-hot sample $x$ such that $x_j = 0$ for $j \neq k$ and $x_k = 1$ with $P(X_k = 1) = \pi_k$. We denote $\text{GM}(\pi)$ to be the distribution whose samples are generated by the Gumbel-Max trick.

A *Gumbel-Softmax* trick (Jang et al., 2017; Maddison et al., 2017) is an alternative to the Gumbel-Max trick that continuously relaxes a categorical random variable. The Gumbel-Softmax utilizes the `softmax` with a temperature $\tau > 0$, instead of the `argmax` in the sampling process, which enables (1) relaxing the discreteness of the categorical random variable to the one-hot-like form $x(\tau) = \texttt{softmax}\left(\frac{\log \pi + g}{\tau}\right)$ in the continuous domain; and (2) approximating the Gumbel-Max by taking $\tau$ small enough. Lately, the Gumbel-Softmax estimator has been widely used to reparameterize categorical random variables, such as `RelaxedOneHotCategorical` in TensorFlow (Abadi et al., 2016). We denote $\text{GS}(\pi, \tau)$ to be the distribution generated by the Gumbel-Softmax trick.

## 3 PROCESS OF GENGS REPARAMETERIZATION

This section discusses the process of GENGS to help understand the concept with minimal theoretical details, and Section 4 provides the theoretical background of GENGS. The three steps of GENGS are the following: (1) approximate a discrete distribution by truncating the distribution; (2) reparameterize the truncated distribution with the Gumbel-Max trick and the linear transformation $\mathcal{T}$, which will be introduced below; and (3) relax the discreteness by replacing the Gumbel-Max trick in Step 2 with the Gumbel-Softmax trick. Figure 1(b) illustrates the full steps of the GENGS trick.

**Step 1.** *Truncate the discrete distribution to finitize the number of possible outcomes.* Suppose $X \sim \text{Poisson}(100)$, which has a mode near at $x = 100$, and near-zero probabilities at $x < 50$ and $x > 150$. The key idea of the first step is ignoring the outcomes of near-zero probabilities at certain levels (ex. $x < 50$ and $x > 150$) and only focusing on the probable samples of meaningful probabilities (ex. $50 \leq x \leq 150$), i.e., truncating the distribution, which finitizes the support of the distribution. Now, suppose we have a discrete random variable $X \sim D(\lambda)$, and its truncated random variable $Z \sim \text{TD}(\lambda, R)$, where $R$ denotes the truncation range that needs to be pre-defined. Proposition 3 in Section 4 provides theoretical reason that $Z$ approximates $X$. Since we finitized the support by the truncation, we may assume $Z$ has a support $C = \{c_0, \cdots, c_{n-1}\}$ of $n$ possible outcomes and its corresponding constant outcome vector $c = (c_0, \cdots, c_{n-1})$. Note that the ordering of $c_k$ is not significant, and Appendix E provides examples of the setting on $c$.

**Step 2.** *Divide sampling process of $Z$ into two-fold: select a one-hot category of $Z$, and revert the selected one-hot category into the original value.* For example, if the sampled value of $Z$ is $c_2 \in C$, we will first focus on the one-hot category class vector $\texttt{one\_hot}(c_2) = (0, 0, 1, 0, \cdots, 0)$, rather than the sampled value $c_2$. Such a one-hot categorical selection process is possible by utilizing the categorical selection $w \sim \text{Categorical}(\pi)$ or its reparameterized version, the Gumbel-Max trick $\text{GM}(\pi)$. Here, the categorical parameter $\pi = (\pi_0, \cdots, \pi_{n-1})$ can be directly calculated by the explicit probability mass funciton (PMF) of the distribution, i.e., $\pi_k = P(Z = c_k)$. However, the PMF of the truncated distribution requires a modification from the PMF of the original distribution, which is determined by how we define $Z$ from $X$. See Definition 1, 2, and Appendix A for detailed configuration of $\pi$. Suppose we now have a one-hot categorical sample $w$ from the categorical parameter $\pi$. Afterward, we revert the one-hot selected categorical vector $w = (w_0, \cdots, w_{n-1})$ into the original sample value with a linear transformation $\mathcal{T}(w) = \sum_k w_k c_k = \sum_k w \odot c$. Proposition 4 shows the validity of the alternative sampling process in Section 4.

**Step 3.** *Relax the one-hot categorical selection into the continuous form by utilizing the Gumbel-Softmax trick.* Up to now, the sole shortage of the reparameterization trick is the *differentiability* due to the *one-hot categorical selecting Gumbel-Max process*. Then, as in Section 2.2, the process can be continuously relaxed with the Gumbel-Softmax trick $\text{GS}(\pi, \tau)$ for some temperature $\tau$. Theorem 5 in Section 4 shows that the alternative sampling process still holds under the continuous relaxation.

## 4 THEORETICAL BACKGROUND OF GENGS

To support our reparameterization methodology, this section provides the main theorem on the reparameterizations. The first proposition approximates an original discrete distribution with its truncated

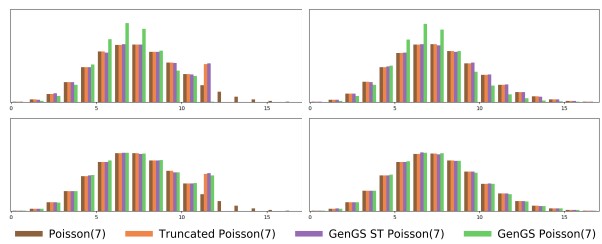 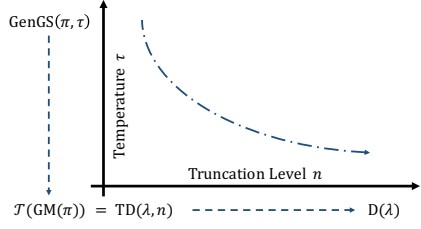

(a) Effects of truncation level and temperature    (b) Concept of GENGS approximation

Figure 2: (a) Approximation of GENGS in terms of choices of the truncation level $n$ and the temperature $\tau$ in Poisson(7). As sub-figures go from left to right, the truncation level grows. Hence, the popped-out sticks, implying remaining probability of the right side, disappears if the truncation level is large enough. As sub-figures go from top to bottom, the temperature decreases, and the PMF of truncated distributions and the original distributions becomes similar. Appendix D provides the fine-grained PMF of GENGS. (b) On the $x$-axis, $\mathrm{TD}(\lambda, n) \to \mathrm{D}(\lambda)$ as truncation level $n \uparrow \infty$, according to Proposition 3. Then, $\mathrm{TD}(\lambda, n)$ can be reparameterized by the Gumbel-Max trick with a linear transformation $\mathcal{T}$ as in Proposition 4. On the $y$-axis, as temperature $\tau \downarrow 0$, $\mathrm{GENGS}(\pi, \tau) \to \mathrm{TD}(\lambda, n)$, where $\pi$ is a computed PMF value of $\mathrm{TD}(\lambda, n)$, according to Theorem 5.

version. Next, the second proposition enables the truncated distribution to be reparameterized by the Gumbel-Max trick. Finally, the main theorem shows that the Gumbel-Softmax trick converges to the Gumbel-Max trick *under an assumption of the linear transformation*. Through these steps, we note that our proposed reparameterization trick is generalized and grounded, theoretically.

### 4.1 FINITIZING THE SUPPORT BY TRUNCATING DISCRETE RANDOM VARIABLES

Definition 1 specifies a *truncated discrete random variable* for truncating the right-hand side. Note that Definition 1 can be easily extended to truncate the left-hand side or both sides of distributions. Definition 2 is truncating both side version, and Appendix A discusses its variation. However, we focus on the non-negative distribution in the remainder of this subsection, since most of the popularly used discrete random variables have the support of $\mathbb{N}_{\geq 0}$.

**Definition 1.** (A special case of right-hand-side truncation for non-negative discrete random variables) A *truncated discrete random variable* $Z_n$ of a non-negative discrete random variable $X \sim \mathrm{D}(\lambda)$ is a discrete random variable such that $Z_n = X$ if $X \leq n - 1$, and $Z_n = n - 1$ if $X > n$. The random variable $Z_n$ is said to follow a *truncated discrete distribution* $\mathrm{TD}(\lambda, R)$ with a parameter $\lambda$ and a truncation range $R = [0, n)$. Alternatively, we write as truncation level $R = n$ if the left truncation is at zero in the non-negative case.

**Definition 2.** (Both-side truncation for general discrete distributions) A *truncated discrete random variable* $Z_{m,n}$ of a discrete random variable $X \sim \mathrm{D}(\lambda)$ is a discrete random variable such that (1) $Z_{m,n} = X$ if $m < X < n$; (2) $Z_{m,n} = n - 1$ if $X \geq n$; and (3) $Z_{m,n} = m + 1$ if $X \leq n$. The random variable $Z_{m,n}$ is said to follow a *truncated discrete distribution* $\mathrm{TD}(\lambda, R = (m, n))$ with a parameter $\lambda$ and a truncation range $R = (m, n)$.

As we discussed in Section 3, truncating the distribution intends to finitize the number of possible outcomes to utilize the categorical selection. From the definition, the samples of finitized support can be simply considered $c_k = k$ in this special non-negative case. Furthermore, due to the definition, the modification on the PMF only exists in the last category $c_{n-1} = n - 1$, and the modified PMF can be computed by injecting the near-zero remaining probability summation to the last category, right before the truncation level. In other words, $\pi_k = P(Z_n = c_k) = P(Z_n = k) = P(X = k)$ for $k = 0, \cdots, n - 2$ and $\pi_{n-1} = 1 - \sum_{k=0}^{n-2} \pi_k$, hence, the *sum-to-one* property remains satisfied. Here, the idea which leads to Proposition 3 is that if we take the truncation level far enough from zero, we can cover most of the possible outcomes that can be sampled from the original distribution. Note that the truncation step can be omitted if the original distribution already has a finite support, but one can utilize the truncation to ignore the unlikely samples.

**Proposition 3.** *With Definition 1, $Z_n$ converges to $X$ almost surely as $n \to \infty$. Also, with Definition 2, $Z_{m,n}$ converges to $X$ almost surely as $m \to -\infty$ and $n \to \infty$.*

The *almost sure convergence* property of Proposition 3 supports the theoretical basis of approximating a discrete random variable $D(\lambda)$ with the truncated random variable $TD(\lambda, n)$, and Appendix A shows the detailed proof. Through the truncation, the discrete distribution is approximated with finitized support, and the Gumbel tricks are ready to be utilized.

### 4.2 REPARAMETERIZATION BY GENERALIZED GUMBEL-SOFTMAX TRICK

Next, we select one-hot categorical sample from the finitized categories and revert the one-hot selection to the original sample value. The widely utilized discrete distributions have the explicit forms of PMF, so we can directly compute the PMF values for the truncated support with a pre-defined truncation range. Once the distribution and the truncation range are fixed as $D(\lambda)$ and $R$, respectively, we have the corresponding constant outcome vector $c = (c_0, \cdots, c_{n-1})$ and the computed PMF value vector $\pi = (\pi_0, \cdots, \pi_{n-1})$ of a truncated distribution $TD(\lambda, R)$, where $\pi_k = TD(c_k; \lambda, R)$. Afterward, we define a transformation $\mathcal{T}(w) = \sum_k w_k c_k = \sum_k w \odot c$. Additionally, we denote the distributions, generated by applying $\mathcal{T}$ on the samples of GM and GS, as $\mathcal{T}(GM)$ and $\mathcal{T}(GS)$, respectively. Then, we can generate a sampled value of $TD(\lambda, R)$ with a linear transformation and a Gumbel-Max sample, as stated in Proposition 4, proved in Appendix B.

**Proposition 4.** *For any truncated discrete random variable $Z \sim TD(\lambda, R)$ of discrete distribution $D(\lambda)$ and a transformation $\mathcal{T}$, $Z$ can be reparameterized by $\mathcal{T}(GM(\pi))$ if we set $\pi_k = P(Z = c_k)$.*

Due to the reparameterization, the randomness of $TD(\lambda, R)$ with respect to the parameter $\lambda$ moves into the uniform random variable in the Gumbel-Max, since $\mathcal{T}$ is a continuous and deterministic function. Then, $TD(\lambda, R)$ can be approximated by replacing the Gumbel-Max with the Gumbel-Softmax in $\mathcal{T}$, as stated in Theorem 5, proved in Appendix C. We define $\mathcal{T}(GS(\pi, \tau))$ as $GENGS(\pi, \tau)$.

**Theorem 5.** *For a transformation $\mathcal{T}$ and a given categorical parameter $\pi \in \Delta^{n-1}$, the convergence property of Gumbel-Softmax to Gumbel-Max still holds under the linear transformation $\mathcal{T}$, i.e., $GS(\pi, \tau) \to GM(\pi)$ as $\tau \to 0$ implies $GENGS(\pi, \tau) \to \mathcal{T}(GM(\pi))$ as $\tau \to 0$.*

Theorem 5 implies that we can relax and approximate the truncated discrete random variable $TD(\lambda, R)$ by the Gumbel-Softmax and the linear transformation. The assumption of the theorem that $GS(\pi, \tau) \to GM(\pi)$ as $\tau \to 0$ has not been mathematically proven in the original literature (Jang et al., 2017; Maddison et al., 2017). Instead, the authors have empirically shown that $GS(\pi, \tau)$ eventually becomes $GM(\pi)$ as $\tau \to 0$. Figure 2 shows how GENGS gets closer to the original distribution by adjusting the truncation range and the temperature.

**Truncation Range.** We can observe that the approximation becomes closer to the original distribution as we widen the truncation range $R$. However, the increment of $R$ is technically limited due to the finite neural network output for the inference. Note that the choice of truncation range is crucial in terms of covering many probable samples. Therefore, we set the truncation range to cover all but less than `1e-5`% of probability with respect to the prior distribution, or arbitrary large.

**Temperature.** The decrement of $\tau$ from $\texttt{softmax}\left(\frac{\log \pi + g}{\tau}\right)$ results in the closer distribution to the original distribution. However, the initially small $\tau$ leads to high bias and variance of gradients, which becomes problematic at the learning stage on $\pi$. Therefore, the annealing of $\tau$ from a large value to a small value is necessary to provide a learning chance of $\pi$.

Note that there is no condition on the distribution to be reparameterized with GENGS in the theoretical analysis. Hence, once a discrete distribution has an explicit PMF, GENGS can be easily utilized to the reparameterization. Appendix E suggests examples of GENGS utilization, including one that shows the regular Gumbel-Softmax is a special case of GENGS. Appendix F provides the algorithm of GENGS, and Appendix G gives discrete distributions in TensorFlow, which can utilize GENGS.

## 5 EXTENTION OF GENGS: IMPLICIT INFERENCE & DISCRETIZATION

**Implicit Inference.** Unlike continuous random variables, discrete random variables have countable outcomes. Instead of inferring the distribution parameter $\lambda$ and then computing the PMF values through the fixed PMF and $\lambda$, we can *directly infer the PMF values $\pi$ of possible outcomes with* `softmax`, which becomes the input of the Gumbel tricks, by loosening the assumption on the approximate posterior PMF shape. This implicit inference on the PMF values becomes possible due to truncating distribution by finitizing the possible outcomes. However, this inference approach needs

a regularizer, such as the KL divergence term in the objective function of VAEs, which ensures the distribution shape to be similar to a prior distribution with a pre-defined parameter. We found that loosening the approximate posterior assumption leads to a significant performance gain in our VAE experiments. See Appendix F for the algorithm of the implicit inference.

**Discretization of Continuously Relaxed Sample.** GENGS outputs a continuously reparameterized sample value since we are relaxing the discrete random variable into a continuous form. Utilizing the Straight-Through (ST) Gumbel-Softmax estimator (Bengio et al., 2013; Jang et al., 2017), instead of the naive Gumbel-Softmax, we can obtain the discrete sample as well. Since ST Gumbel-Softmax discretizes the relaxed Gumbel-Softmax output with `argmax`, ST Gumbel-Softmax uses the gradients obtained from the relaxed ones, which could result in significant performance degradation.

## 6 RELATED WORK

GENGS is basically a single-sample gradient estimator like other reparameterization gradient estimators. Though GENGS could use multiple samples to obtain the stable gradients, we compare GENGS with the other estimators using a single sample to test the fundamental performance of gradient estimators. RF denotes the basic REINFORCE (Williams, 1992). NVIL (Mnih & Gregor, 2014) utilizes a neural network to introduce the optimal control variate. MUPROP (Gu et al., 2016) utilizes the first-order Taylor expansion on the loss term as a control variate. VIMCO($k$) (Mnih & Rezende, 2016) is designed as $k$-sample gradient estimator. REBAR (Tucker et al., 2017) and RELAX (Grathwohl et al., 2017) utilize reparameterization trick for constructing the control variate. Deterministic RaoBlack (DETRB) (Liu et al., 2019) uses the weighted value of the fixed gradients from $m$-selected categories and the estimated gradients from the remaining categories with respect to their odds to reduce the variance. The idea of Stochastic RaoBlack (STORB) (Kool et al., 2020) is essentially same as that of DETRB, but STORB randomly chooses the categories at each step, instead of using fixed categories. Kool et al. (2020) also suggested an unordered set gradient estimator (UNORD), which also uses the multiple sampled gradients, utilizing the sampling without replacements. For DETRB, STORB, and UNORD, we use $m = 1$ category that utilizes the fixed gradient for the fair comparison. Note that if there are $K > 1$ dimensions to be inferred, the models require computing $m^K$ gradient combinations, which has higher complexity than GENGS. The $^*$ symbol denotes a variation that utilizes a built-in control variate introduced in the work of Kool et al. (2020).

## 7 EXPERIMENT

### 7.1 SYNTHETIC EXAMPLE

**Experimental Setting.** In this experiment, we expand the toy experiments from Tucker et al. (2017); Grathwohl et al. (2017) to various discrete distributions. We first fix constant $t_1, \cdots, t_k$, and then optimize the loss function $\mathbb{E}_{z \sim p(z|\lambda)}\left[\sum_{i=1}^{k}(z_i - t_i)^2\right]$ with respect to $\lambda$. Here, we set $p(z|\lambda)$ as Poisson(20), Binomial(20, .3), Multinomial(3, [.7, .2, .1]), and NegativeBinomial(3, .4) in this experiment. We also adapt the Rao-Blackwellization (RB) idea in GENGS by utilizing $m = 1$ in calculating the selected gradient, so this adaptation results in GENGS-RB that estimates the remaining gradients by GENGS. See Appendix J for the detailed experimental settings.

**Experimental Result.** Figure 3 compares the log-loss and the log-variance of estimated gradients from various estimators. In the figure, the log-loss needs to be minimized to correctly estimate the backpropagated gradient value in the learning process. Additionally, the log-variance requires being minimized to maintain the consistency of the gradients, so the gradient descent can be efficient. GENGS shows the best log-loss and the best log-variance if GENGS keeps the continuous relaxation of the modeled discrete random variable. For the Poisson case, the exact gradient has a closed-form solution, as in Appendix J, and GENGS shows the lowest bias among all gradient estimators. See Appendix J for the curves with confidence intervals and the curves without smoothing.

### 7.2 VAE: SYNTHETIC EXPERIMENT ON DEEP GENERATIVE MODELS

**Experimental Setting.** We follow the VAE experiments of Figurnov et al. (2018) with discrete priors, which diversifies the choice of prior assumption, as the *latent factor count* in the discrete case. This

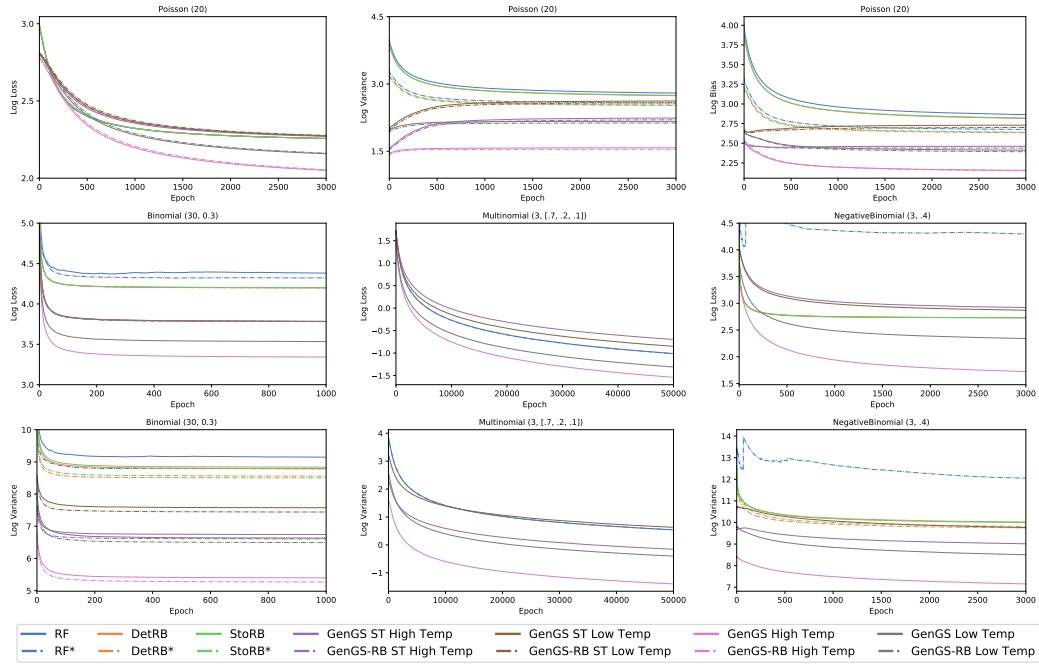

Figure 3: Synthetic example performance curves in log scale: (Top Row) losses, variances, and biases of gradients for Poisson; (Middle Row) losses for Binomial, Multinomial, and NegativeBinomial; (Bottom Row) variances of gradients for Binomial, Multinomial, and NegativeBinomial. We utilize the cumulative average for smoothing the curves, and the curves with confidence intervals and the curves without smoothing are in Appendix J.

Table 1: Test negative ELBO on MNIST and OMNIGLOT datasets. The lower is better for the negative ELBO. We provide a full table including baselines with insignificant results and variations of GENGS in Appendix K. Symbol "—" indicates no convergence.

| MNIST | RF* | NVIL | MUPROP | VIMCO(5) | REBAR | RELAX | STORB* | GENGS (Ex.) | GENGS (Im.) |
|---|---|---|---|---|---|---|---|---|---|
| Pois(2) | $122.81_{\pm 2.41}$ | $129.34_{\pm 4.72}$ | $125.43_{\pm 2.27}$ | $122.55_{\pm 3.28}$ | $123.44_{\pm 2.54}$ | $122.71_{\pm 1.92}$ | $124.02_{\pm 4.91}$ | $103.18_{\pm 0.92}$ | $96.04_{\pm 1.44}$ |
| Pois(3) | $123.12_{\pm 2.21}$ | $130.24_{\pm 3.32}$ | $125.92_{\pm 1.81}$ | $121.15_{\pm 2.57}$ | $120.62_{\pm 2.31}$ | $119.84_{\pm 2.18}$ | $124.41_{\pm 5.96}$ | $105.15_{\pm 1.71}$ | $96.01_{\pm 1.27}$ |
| Geom(.25) | $127.90_{\pm 1.97}$ | $135.90_{\pm 2.38}$ | $137.90_{\pm 2.14}$ | $127.21_{\pm 2.55}$ | $135.12_{\pm 2.74}$ | $136.80_{\pm 3.06}$ | $131.09_{\pm 4.95}$ | $98.43_{\pm 0.81}$ | $92.52_{\pm 1.62}$ |
| Geom(.5) | $129.20_{\pm 2.03}$ | $138.47_{\pm 2.30}$ | $136.40_{\pm 1.78}$ | $129.91_{\pm 2.90}$ | $138.37_{\pm 2.98}$ | $139.41_{\pm 3.59}$ | $139.67_{\pm 2.42}$ | $100.92_{\pm 1.24}$ | $93.81_{\pm 1.60}$ |
| NegBin(3,.5) | $116.67_{\pm 5.97}$ | $119.28_{\pm 7.80}$ | $131.96_{\pm 6.49}$ | $112.69_{\pm 4.30}$ | — | — | $114.36_{\pm 4.12}$ | $98.58_{\pm 1.27}$ | $94.52_{\pm 1.52}$ |
| NegBin(5,.3) | $130.03_{\pm 3.99}$ | $133.44_{\pm 4.27}$ | $144.05_{\pm 8.15}$ | $124.48_{\pm 2.72}$ | — | — | $128.02_{\pm 2.60}$ | $100.88_{\pm 2.35}$ | $95.37_{\pm 1.43}$ |
| OMNIGLOT | RF* | NVIL | MUPROP | VIMCO(5) | REBAR | RELAX | STORB* | GENGS (Ex.) | GENGS (Im.) |
| Pois(2) | $139.47_{\pm 3.29}$ | $148.01_{\pm 4.19}$ | $142.95_{\pm 1.32}$ | $138.73_{\pm 3.42}$ | $138.12_{\pm 3.26}$ | $137.56_{\pm 2.94}$ | $139.61_{\pm 5.87}$ | $127.89_{\pm 1.44}$ | $118.17_{\pm 2.22}$ |
| Pois(3) | $140.54_{\pm 2.36}$ | $148.13_{\pm 3.98}$ | $143.85_{\pm 1.54}$ | $139.37_{\pm 3.10}$ | $137.92_{\pm 3.07}$ | $137.42_{\pm 2.96}$ | $140.05_{\pm 3.68}$ | $131.53_{\pm 1.76}$ | $119.15_{\pm 1.92}$ |
| Geom(.25) | $142.68_{\pm 2.96}$ | $153.69_{\pm 2.52}$ | $152.17_{\pm 1.77}$ | $142.94_{\pm 3.96}$ | $146.78_{\pm 3.62}$ | $148.91_{\pm 4.03}$ | $143.10_{\pm 3.91}$ | $115.23_{\pm 2.00}$ | $107.79_{\pm 2.84}$ |
| Geom(.5) | $142.70_{\pm 1.77}$ | $153.20_{\pm 1.49}$ | $149.76_{\pm 2.19}$ | $142.05_{\pm 3.56}$ | $149.63_{\pm 3.49}$ | $151.97_{\pm 3.90}$ | $142.56_{\pm 2.97}$ | $115.14_{\pm 2.43}$ | $108.48_{\pm 2.78}$ |
| NegBin(3,.5) | $141.44_{\pm 2.20}$ | $144.44_{\pm 2.78}$ | $147.78_{\pm 4.49}$ | $141.89_{\pm 3.84}$ | — | — | $129.48_{\pm 4.34}$ | $118.57_{\pm 2.71}$ | $117.02_{\pm 2.18}$ |
| NegBin(5,.3) | $144.44_{\pm 3.68}$ | $159.40_{\pm 5.13}$ | $152.81_{\pm 3.34}$ | $150.49_{\pm 4.09}$ | — | — | $151.30_{\pm 3.98}$ | $119.57_{\pm 2.02}$ | $117.54_{\pm 2.76}$ |

experiment utilizes the Poisson, the geometric, and the negative binomial distributions. The evidence lower bound (ELBO) $\mathcal{L}(\mathbf{x}) = \mathbb{E}_{q_{\phi(\mathbf{z}|\mathbf{x})}}[\log p_\theta(\mathbf{x}|\mathbf{z})] - \text{KL}(q_\phi(\mathbf{z}|\mathbf{x})||p_\theta(\mathbf{z}))$, which consists of the reconstruction part and the KL divergence part, is minimized during the training period. Optimizing the ELBO of VAEs requires computing the KL divergence between the approximate posterior and the prior distributions. In GENGS, by truncating the original distribution, the KL divergence becomes the derivation with categorical distributions. See Appendix H for the detailed statement and proof.

Note that the purpose of VAE experiments is not to compare the performance across various prior distributions. The VAE is considered as a more challenging task than the synthetic example in the last subsection, since (1) this task requires computing the gradients of the encoder network parameters through the latent distribution parameter $\lambda$; and (2) each stochastic gradient of the latent dimension affects every encoder parameter since we are utilizing the fully-connected layers. Hence, a single poorly estimated gradient with respect to the latent distribution parameter $\lambda$ could negatively affect the learning of encoder parameters. See Appendix K for the detailed experimental settings.

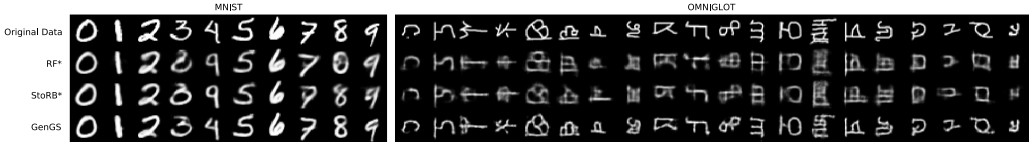

Figure 4: Reconstructed images by VAEs with various gradient estimators. GENGS shows the clearest images among other gradient estimators with better reconstruction.

**Experimental Result.** Table 1 shows the negative ELBO results of the VAE experiments. We found that some baselines failed to reach the optimal point, so we excluded those estimators in such suboptimal cases. The variants of GENGS showed the lowest negative ELBO in general, and loosening the PMF condition idea (i.e., the implicit inference) reached the better optimal points. Figure 4 shows the reconstructed images by VAEs with various gradient estimators on MNIST and OMNIGLOT. GENGS draws the clearest images and better reconstructions, which aligns with the quantitative result in Table 1. See Appendix K for the full table and additional discussion.

### 7.3 TOPIC MODEL APPLICATION

**Experimental Setting.** This experiment shows another application of GENGS in the topic modeling. The Poisson distribution is one of the most important distribution for counting the number of outcomes among all discrete distributions. The authors of *Deep Exponential Families* (DEFs) (Ranganath et al., 2015) utilized the exponential family, including the Poisson distribution, on the stacked latent layers. Therefore, we focus on the Poisson DEF, which assumes the Poisson latent layers to capture the counting

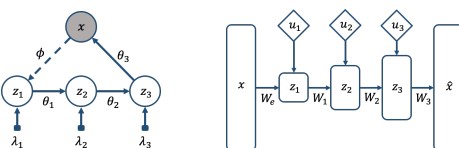

Figure 5: (Left) A graphical notation of NVPDEF with generative process ($\theta$) and inference network ($\phi$). The multi-stacked latent layers have $\lambda_i$ as a prior distribution parameter. (Right) A neural network diagram of NVPDEF: diamond nodes indicate the auxiliary random variable for the reparameterization trick.

numbers of latent super-topics and sub-topics; and we convert the Poisson DEF into a neural variational form, which resembles NVDM (Miao et al., 2016). Figure 5 shows the neural network and its corresponding probabilistic modeling structure. We utilize GENGS on the Poisson DEF to sample the values in the latent variable, namely the neural variational Poisson DEF (NVPDEF). See Appendix L for further description of DEFs, NVPDEF, and detailed experimental settings.

**Experimental Result.** We enumerate the baselines and the variants of NVPDEFs in Table 2, and we confirmed that NVPDEF shows the lowest perplexity overall with 20Newsgroups and RCV1-V2. Since NVPDEF and the original DEFs have different training and testing regimes, we compare NVPDEF to representative neural variational topic (document) models, which are listed in Table 2. Additionally, Appendix L shows the qualitative results from topic models.

Table 2: Test perplexity on 20News and RCV dataset.

| Model | 20News (Dim.) | RCV (Dim.) |
|---|---|---|
| LDA (Blei et al., 2003) | $1082_{\pm 12.9}$ (50) | $1187_{\pm 15.4}$ (200) |
| NVDM (Miao et al., 2016) | $803_{\pm 9.3}$ (50) | $574_{\pm 18.3}$ (200) |
| GSM (Miao et al., 2017) | $854_{\pm 7.1}$ (50) | $801_{\pm 5.2}$ (200) |
| NVLDA (Srivastava & Sutton, 2017) | $1155_{\pm 16.5}$ (50) | $1574_{\pm 24.7}$ (200) |
| PRODLDA (Srivastava & Sutton, 2017) | $1145_{\pm 13.3}$ (50) | $1425_{\pm 17.1}$ (200) |
| NVPDEF | $759_{\pm 13.1}$ (50) | $562_{\pm 11.5}$ (200) |
| MULTI-STACKED NVPDEF | $783_{\pm 17.6}$ (20-50) | $576_{\pm 18.8}$ (50-200) |

## 8 CONCLUSION

This paper suggests a new gradient estimator for general discrete random variables, a generalized version of the Gumbel-Softmax estimator. To strengthen the practical usage of reparameterization tricks with the Gumbel-Softmax function, we provide a theoretical background of our reparameterization trick. Our finding claims that a discrete random variable can always be reparameterized through the proposed GENGS algorithm. The limitation of GENGS is the setting of the truncation level and the Gumbel-Softmax temperature, which becomes the trade-off between the gradient estimation accuracy and the time budget. Subsequently, we show the synthetic analysis and the VAE experiment, as well as topic model application of GENGS. With the generalization, we expect that GENGS can diversify the options of distributions in the deep generative model community.

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

## A    PROOF OF PROPOSITION 3 & TRUNCATING BOTH SIDES

### A.1    PROOF OF PROPOSITION 3 FOR TRUNCATING RIGHT-HAND-SIDE

Before reminding Definition 1 and Proposition 3 of the main paper, we first introduce the definition of *almost sure convergence* of sequence of random variables.

**Definition.** (Almost Sure Convergence) For a sequence of random variables $\{X_n\}_{n=1}^{\infty}$, $X_n$ *converges almost surely* to random variable $X$, if

$$P(\lim_{n \to \infty} X_n = X) = P(\{\omega \in \Omega | \lim_{n \to \infty} X_n(\omega) = X(\omega)\}) = 1 \qquad (2)$$

for a sample space $\Omega$.

**Definition.** (A special case of right-hand-side truncation for non-negative discrete random variables) A *truncated discrete random variable* $Z_n$ of a non-negative discrete random variable $X \sim D(\lambda)$ is a discrete random variable such that $Z_n = X$ if $X \leq n - 1$, and $Z_n = n - 1$ if $X > n$. The random variable $Z_n$ is said to follow a *truncated discrete distribution* $TD(\lambda, R)$ with a parameter $\lambda$ and a truncation range $R = [0, n)$. Alternatively, we write as truncation level $R = n$ if the left truncation is at zero in the non-negative case.

With this definition, as we discussed in the main text, the constant outcome vector can be defined as $c = (0, 1, 2, \cdots, n - 1)$. Note that for $Z_n = 0, 1, \cdots, n - 2$, the sample spaces of $Z_n$ and $X$ are equal, hence, $P(Z_n) = P(X)$ in such cases. This implies that the modified PMF of $Z_n = k$ can be computed by using the PMF of $X = k$[1] in such cases, i.e., $P(Z_n = k) = P(X = k)$ for $k = 0, 1, \cdots, n-2$. Then, consequently, $P(Z_n = n-1) = P(X \geq n-1) = 1 - \sum_{k=0}^{n-2} P(Z_n = k)$ due to the sum-to-one property of probability. Hence, the corresponding categorical parameter $\pi$ of the constant outcome vector $c$ can be computed using the PMF of the original discrete random variable $X$: $\pi_k = P(Z_n = k) = P(X = k)$ for $k = 0, 1, \cdots, n-2$, and $\pi_{n-1} = P(Z_n = n-1) = P(X \geq n - 1) = 1 - \sum_{k=0}^{n-2} \pi_k$.

**Proposition.** *With Definition 1, $Z_n$ converges to $X$ almost surely as $n \to \infty$.*

*Proof.* Note that $\{\omega \in \Omega | Z_n(\omega) = X(\omega)\} = \{\omega \in \Omega | X(\omega) < n\}$. Then, we have the following:

$$P(\lim_{n \to \infty} Z_n = X) = P(\{\omega \in \Omega | \lim_{n \to \infty} Z_n(\omega) = X(\omega)\}) \qquad (3)$$

$$= P(\{\omega \in \Omega | \lim_{n \to \infty} Z_n(w) = \lim_{n \to \infty} X(w) < \lim_{n \to \infty} n = \infty\}\}) \qquad (4)$$

$$= P(\{\omega \in \Omega | X(\omega) < \infty\}) \qquad (5)$$

$$= P(X < \infty) \qquad (6)$$

$$= 1 \qquad (7)$$

since $X$ is a non-negative discrete random variable, and its cumulative distribution function $F_X(x)$ has the following properties: non-decreasing as $x \to \infty$, and $0 \leq F_X \leq 1$. $\qquad \square$

### A.2    PROPOSITION 3 FOR TRUNCATING BOTH SIDES

The below is truncating both side version of Definition 2 and Proposition 3 in the main paper. For simplicity, we assume the distribution has support $\mathbb{Z}$.

**Definition.** (Both-side truncation for general discrete distributions) A *truncated discrete random variable* $Z_{m,n}$ of a discrete random variable $X \sim D(\lambda)$ is a discrete random variable such that

$Z_{m,n} = \begin{cases} X, & \text{if } m < X < n \\ n - 1, & \text{if } X \geq n \\ m + 1, & \text{if } X \leq m \end{cases}$ . The random variable $Z_{m,n}$ is said to follow a *truncated*

*discrete distribution* $TD(\lambda, R = (m, n))$ with a parameter $\lambda$ and a truncation range $R = (m, n)$.

With this definition, the constant outcome vector can be defined as $c = (m + 1, \cdots, n - 1)$. Note that for $Z_{m,n} = m + 2, \cdots, n - 2$, the sample spaces of $Z_{m,n}$ and $X$ are equal, hence,

---

[1]Note that we are using the distribution with eplicitly known PMF for discrete distribution $X$.

$P(Z_{m,n}) = P(X)$ in such cases. This implies that the modified PMF of $Z_{m,n} = k$ can be computed by using the PMF of $X = k$ in such cases, i.e., $P(Z_{m,n} = k) = P(X = k)$ for $k = m + 2, \cdots, n - 2$. Then, with the definition, $P(Z_{m,n} = m + 1) = P(X \leq m + 1)$ and $P(Z_{m,n} = n - 1) = P(X \geq n - 1)$. However, during the implementation, this definition can be a problem since the configuration implies that there might be a case that we have to compute two infinite sums, which are $P(Z_{m,n} = m + 1) = P(X \leq m + 1) = \sum_{k \leq m+1} P(X = k)$ and $P(Z_{m,n} = n - 1) = P(X \geq n - 1) = \sum_{k \geq n-1} P(X = k)^2$. Hence, we also provide an alternative configuration of $Z_{m,n}$ in the later of this section.

**Proposition.** *With definition above for the truncating both sides, $Z_{m,n}$ converges to $X$ almost surely as $m \to -\infty$ and $n \to \infty$.*

*Proof.* Note that $\{\omega \in \Omega | Z_{m,n}(\omega) = X(\omega)\} = \{\omega \in \Omega | m < X(\omega) < n\}$. Then, we have the following:

$$P(\lim_{\substack{n \to \infty \\ m \to -\infty}} Z_{m,n} = X) = P(\{\omega \in \Omega | \lim_{\substack{n \to \infty \\ m \to -\infty}} Z_{m,n}(\omega) = X(\omega)\}) \tag{8}$$

$$= P(\{\omega \in \Omega | -\infty = \lim_{m \to -\infty} m < X(w) < \lim_{n \to \infty} n = \infty\}\}) \tag{9}$$

$$= P(\{\omega \in \Omega | -\infty < X(\omega) < \infty\}) \tag{10}$$

$$= P(-\infty < X < \infty) \tag{11}$$

$$= P(X < \infty) - P(-\infty < X) \tag{12}$$

$$= 1 \tag{13}$$

since $X$ is a discrete random variable, and its cumulative distribution function $F_X(x)$ has the following properties: non-decreasing as $x \to \infty$, non-increasing as $x \to -\infty$, and $0 \leq F_X \leq 1$. $\square$

As we discussed above, during the implementation, computing the small probabilities of both left and right tail can cause a problem, either it can have high complexity[3] or even impossible[4]. Hence, when we can consider the alternative definition such as $Z_{m,n} = \begin{cases} X, & \text{if } m < X < n \\ n - 1, & \text{if } X \geq n \text{ or } X \leq m \end{cases}$, which has a simpler PMF computation. In this case, we can simply add the remaining probability of unlikely samples at the right-most value, and moreover, the alternative configuration does not harm the proof of the proposition. Hence, with the alternative defintion, the the corresponding categorical parameter $\pi = (\pi_{m+1}, \cdots, \pi_{n-1})$ of the constant outcome vector $c = (m + 1, \cdots, n - 1)$ can be computed using the PMF of the original discrete random variable $X$: $\pi_k = P(Z_n = k) = P(X = k)$ for $k = m + 1, \cdots, n - 2$, and $\pi_{n-1} = P(Z_n = n - 1) = P(X \geq n - 1) + P(X < m + 1) = 1 - \sum_{k=m+1}^{n-2} \pi_k$.

## B  PROOF OF PROPOSITION 4

**Proposition.** *For any truncated discrete random variable $Z \sim TD(\lambda, R)$ of discrete distribution $D(\lambda)$ and a transformation $\mathcal{T}$, $Z$ can be reparameterized by $\mathcal{T}(GM(\pi))$ if we set $\pi_k = P(Z = c_k)$.*

*Proof.* Note that $Z$ has two parameters: the distribution parameter $\lambda$ and the truncation range $R$. Assume that we have possible outcome set $C = \{c_0, \cdots, c_{n-1}\}$ of $n$ possible outcomes by truncating the distribution with truncation range $R$. Note that the transformation is defined as $\mathcal{T}(w) = \sum_{k=0}^{n-1} w_k c_k = \sum_k w \odot c$. By pre-defining the truncation range $R$ as a hyper-parameter, the randomness of $Z$ is fully dependent on the distribution parameter $\lambda$. Now, we introduce the Gumbel random variable $G = -\log(-\log U)$ where $U \sim \text{Uniform}(0, 1)$ as an auxiliary random variable. Then, given a categorical parameter $\pi \in \Delta^{n-1}$, any $n$-dimensional one-hot vector $e_j = $

---

[2]Or, there might be a case that one of $P(Z_{m,n} = m + 1) = P(X \leq m + 1)$ and $P(Z_{m,n} = n - 1) = P(X \geq n - 1)$ requires summation of high complexity, even though it is a finite summation.

[3]For example, if we truncate $\text{Poisson}(1000)$ with truncation range $R = [900, 1100]$, we have to compute PMF values for $0 \leq x < 900$ to sum-up the left-hand meaningless probability, and it causes high complexity.

[4]For example, the case when the support has infinite support for both left ($-\infty$) and right ($\infty$) sides.

$(0, \cdots, 0, 1, 0, \cdots, 0)$, which has 1 in $j^{\text{th}}$ entry and 0 in all other entries, can be reparameterized by Gumbel-Max trick.

Suppose we have a sample $c_m$ from $Z$, and note that we have known PMF values $\pi = (\pi_0, \cdots, \pi_{n-1})$ of $Z$ by the definition of $Z$. Then, with the transformation $\mathcal{T}$ and the constant outcome vector $c = (c_0, \cdots, c_{n-1})$, the following holds:

$$c_m = \sum_{k=0}^{n-1} c_k e_k = \mathcal{T}(e_m) . \tag{14}$$

Since the transformation $\mathcal{T}$ is also a deterministic function, by introucding the Gumbel random variable as an auxiliary random variable, we can replace the randomness of $Z$ from $\lambda$ (or $\pi$ in the implicit inference case) with the uniform random variable composing the Gumbel random variable. Hence, the truncated discrete random variable $Z$ can be reparameterized by the Gumbel-Max trick and the transformation $\mathcal{T}$. □

## C  PROOF OF THEOREM 5

**Theorem.** *For a transformation $\mathcal{T}$ and a given categorical parameter $\pi \in \Delta^{n-1}$, the convergence property of Gumbel-Softmax to Gumbel-Max still holds under the linear transformation $\mathcal{T}$, i.e., $GS(\pi, \tau) \to GM(\pi)$ as $\tau \to 0$ implies $\text{GENGS}(\pi, \tau) \to \mathcal{T}(GM(\pi))$ as $\tau \to 0$.*

*Proof.* Suppose we have given a categorical parameter $\pi \in \Delta^{n-1}$. Define $f_M$ be a Gumbel-Max trick function, and $f_S^\tau$ be a Gumbel-Softmax trick function with a temperature $\tau > 0$ that both functions take the categorical parameter and a Gumbel sample as inputs. Note that $f_M$ returns a one-hot vector which has 1 in the `argmax` entry after the Gumbel perturbation, and $f_S^\tau$ returns a one-hot-like `softmax` activation value with the temperature $\tau$ with the Gumbel perturbation.

Draw a random sample $u \sim \text{Uniform}(0, 1)^n$ which defines the Gumbel sample $g$ for the Gumbel perturbation. Assume that $\log \pi_m - \log(-\log u_m) > \log \pi_j - \log(-\log u_j)$ for all $j \neq m$, i.e., the selected sample as a category is $c_m$ out of possible outcome set $\{c_0, \cdots, c_{n-1}\}$. Therefore, for the Gumbel-Max trick, it is clear that $\mathcal{T}(f_M(\pi, g)) = \sum_{k=0}^{n-1} [e_m]_k \cdot c_k = \sum_{k=0}^{n-1} e_m \odot c = c_m$ for $c = (c_0, \cdots, c_{n-1})$ where $e_j = (0, \cdots, 0, 1, 0, \cdots, 0)$ is a $n$-dimensional one-hot vector, which has 1 in the $m^{\text{th}}$ entry and 0 in all other entries. Then, the statement that $GS(\pi, \tau) \to GM(\pi)$ as $\tau \to 0$ implies $f_S^\tau(\pi, g) \to f_M(\pi, g)$, i.e., the following:

$$\left[ f_S^\tau(\pi, g) \right]_j = \frac{\exp \left( \frac{\log \pi_j - \log(-\log u_j)}{\tau} \right)}{\sum_{k=0}^{n-1} \exp \left( \frac{\log \pi_k - \log(-\log u_k)}{\tau} \right)} \tag{15}$$

$$\to \begin{cases} 1 \text{ if } j = m \\ 0 \text{ if } j \neq m \end{cases} \quad \text{as } \tau \to 0 . \tag{16}$$

Then, $f_S^\tau(\pi, g) = \tilde{e}_m$ for some relaxed one-hot vector of $e_m$ by introducing the softmax relaxation. As a consequence,

$$\left[ f_S^\tau(\pi, g) \right]_j \times c_j = \left[ \sum \frac{\exp \left( \frac{\log \pi_i - \log(-\log u_i)}{\tau} \right)}{\sum_{k=0}^{n-1} \exp \left( \frac{\log \pi_k - \log(-\log u_k)}{\tau} \right)} \right]_j \times c_j \tag{17}$$

$$\to \begin{cases} c_m \text{ if } j = m \\ 0 \quad \text{if } j \neq m \end{cases} \quad \text{as } \tau \to 0 , \tag{18}$$

since the constant multiplication gives no harm to the approximation. Hence, by taking the summation, which also gives no harm to the approximation, $\mathcal{T}(f_S^\tau(\pi, g)) = \sum_{k=0}^{n-1} [\tilde{e}_m]_k \cdot c_k \to c_m = \sum_{k=0}^{n-1} \tilde{e}_m \odot c \to c_m = \mathcal{T}(f_M(\pi, g))$. □

## D  PMF SHAPE OF GENGS

Note that Figure 2(b) in the main paper is drawn by rounding-up continuous values into integers. Since PMF for discrete distributions (Poisson(7) in Figure 2(b)) and PDF for continuous distributions (GENGS for Poisson(7) in Figure 2(b)) cannot be directly compared within the same figure due to their scale, we provide Figure 6 of the fine-grained PMF by rounding-up in small decimals.

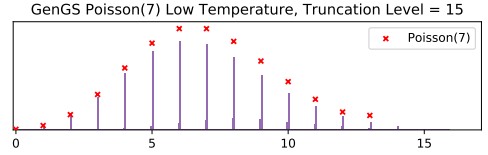 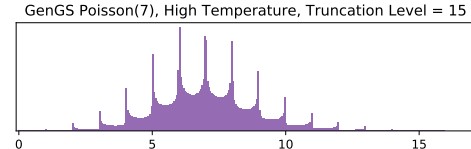

Figure 6: Fine-grained PMF of GenGS for Poisson$(7)$.

# E    EXAMPLES OF GENGS

## E.1    TRUNCATING RIGHT-HAND SIDE FOR POISSON DISTRIBUTION

For the distributions that the left-side truncation needs to be at zero, such as the Poisson with small rate parameter, the geometric, the negative binomial, etc., as we mentioned in the main text, we can simply set the constant outcome vector $c = (0, 1, \cdots, n - 1)$. Note that the ordering of $c$ is not crucial, for example, we can also set $c = (2, 1, 0, 3, 4, \cdots, n - 1)$. Then, the corresponding PMF value needs to be computed as $\pi = \big(P(Z = 2), P(Z = 1), P(Z = 0), P(Z = 3), P(Z = 4), \cdots, P(Z = n-1)\big)$, where $Z$ is the truncated discrete random variable.

## E.2    TRUNCATING BOTH SIDES FOR POISSON DISTRIBUTION

For the case when the distribution requires left-side truncation, for example, Poisson$(100)$[5], we can set the constant outcome vector such as $c = (50, 51, \cdots, 149, 150)$. Again, the ordering of $c$ is not crucial, hence, we can also set $c = (148, 150, 149, 147, \cdots, 53, 50, 52, 51)$, for example. Afterward, the PMF value $\pi$ is naturally computed in the same order as the constant outcome vector $c$.

## E.3    GUMBEL-SOFTMAX IS A SPECIAL CASE OF GENGS.

The Gumbel-Softmax estimator of categorical random variables is a trivial case of GENGS. Assume the number of dimensions $n = 3$ and a categorical parameter $\pi = (0.5, 0.3, 0.2)$ in this example. Then, the poissble outcomes of Categorical$(\pi)$ are $c_0 = [1, 0, 0]^T$, $c_1 = [0, 1, 0]^T$, and $c_2 = [0, 0, 1]^T$. Afterward, draw a sample $w$ from GS$(\pi, \tau)$ for some temperature $\tau > 0$, and the value will become a relaxed one-hot form, for example, $(0.95, 0.04, 0.01)$. If we construct

$$c = [c_0, c_1, c_2] \tag{19}$$

$$= \begin{bmatrix} 1 & 0 & 0 \\ 0 & 1 & 0 \\ 0 & 0 & 1 \end{bmatrix} = I_3 \, , \tag{20}$$

then $\mathcal{T}(w) = \sum_k w \odot c = \sum_k w_k c_k \approx w$. Hence, the Gumbel-Softmax trick can be written in the form of GENGS with the identity matrix in the linear transformation. Note that we abuse the symbol of Hadamard product ($\odot$) in terms of the dimension, where the last term of $\mathcal{T}(w) = \sum_k w \odot c = \sum_k w_k c_k$ is actually the multiplication of a scalar $w_k \in [0, 1]$ and a vector $c_k \in \mathbb{R}^3$ in this case.

## E.4    GENGS CAN BE APPLIED TO MULTINOMIAL DISTRIBUTION.

If we go one step forward from the example above, we can reparameterize multinomial distribution with GENGS trick, also. For example, assume the number of trial $m = 3$ and the probability vector $p = [p_1, p_2, p_3] = [.7, .2, .1]$. Then, the possible outcomes are $c_0 = [3, 0, 0]^T$, $c_1 = [0, 3, 0]^T$, $c_2 = [0, 0, 3]^T$, $c_3 = [2, 1, 0]^T$, $c_4 = [2, 0, 1]^T$, $c_5 = [1, 2, 0]^T$, $c_6 = [1, 0, 2]^T$, $c_7 = [0, 2, 1]^T$, $c_8 = [0, 1, 2]^T$, and $c_9 = [1, 1, 1]^T$, where the corresponding probability is $\frac{(n_1+n_2+n_3)!}{n_1!n_2!n_3!} p_1^{n_1} p_2^{n_2} p_3^{n_3}$ for

---

[5]Note that Poisson$(100)$ has inprobable samples at $x < 50$ and $x > 150$.

outcome $[n_1, n_2, n_3]$. Construct the linear transformation constant $c$ as the following:

$$c = [c_0 \ c_1 \ c_2 \ c_3 \ c_4 \ c_5 \ c_6 \ c_7 \ c_8 \ c_9] \tag{21}$$

$$= \begin{bmatrix} 3 & 0 & 0 & 2 & 2 & 1 & 1 & 0 & 0 & 1 \\ 0 & 3 & 0 & 1 & 0 & 2 & 0 & 2 & 1 & 1 \\ 0 & 0 & 3 & 0 & 1 & 0 & 2 & 1 & 2 & 1 \end{bmatrix}, \tag{22}$$

where the categorical parameter $\pi = \left( \frac{(n_1+n_2+n_3)!}{n_1!n_2!n_3!} p_1^{n_1} p_2^{n_2} p_3^{n_3} \right)_{[n_1, n_2, n_3]}$. If we sample a Gumbel-Softmax sample $w$ from $GS(\pi, \tau)$ and compute $\mathcal{T} = \sum_k w \odot c$, the result will be in the relaxed form of selected samples $c_m$. This example shows how the linear transformation constant $c$ can be generalized to the matrix form. However, if we recall that the equation $n_0 + \cdots + n_{k-1} = n$ has $\binom{n+k-1}{k}$ solutions of non-negative tuple $(n_0, \cdots, n_{k-1})$, where $\binom{n+k-1}{k} \leq \mathcal{O}(n^{\min\{k, n-1\}})$, the relaxed categorical selection through the Gumbel-Softmax can become problematic due to the high complexity when $n$ or $k$ has large value. Hence, in this situation, reducing the possible outcomes by disregarding unlikely samples by user guidance can be a remedy, but the treatment can not be a fundamental solution, and handling such case can be an open research question.

## F  INFERENCE STEP, ALGORITHM & COMPLEXITY OF GENGS

### F.1  VISUALIZATION OF GENGS REPARAMETRIZATION STEPS

Figure 7 and Figure 8 represent the reparameterization stpes of explicit inference and implicit inference of GENGS, respectively. For both figures, the shaded nodes indicate the auxiliary Gumbel samples, composed of uniform samples, which enable the reparameterized variable to be deterministic with respect to the parameter of the target distribution.

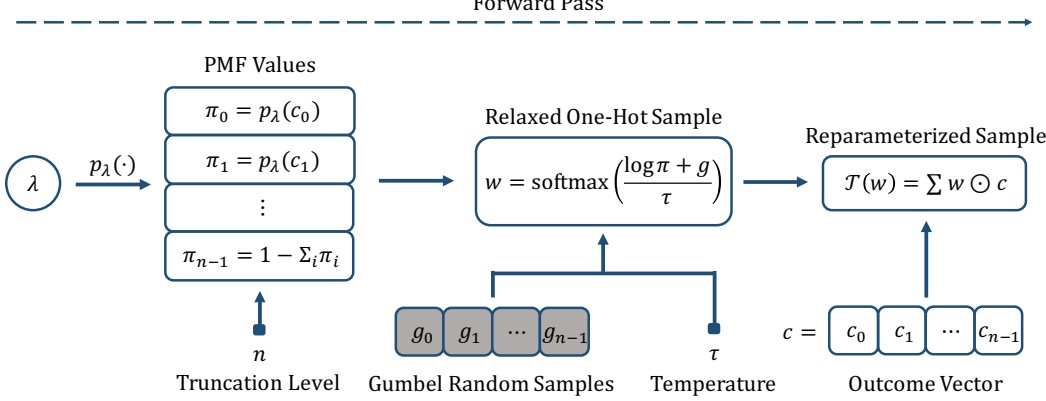

Figure 7: Visualization of GENGS reparameterization of explicit inference version. Note that we compute PMF values from the infered parameter $\lambda$.

### F.2  ALGORITHM & ADDITIONAL COMPUTATIONAL COMPLEXITY OF GENGS

Algorithm 1 and Algorithm 2 provides the algorithm of explicit and implicit inference steps of GENGS, respectively. The explicit inference has further computation on PMF value calculation in Line 3-4 of Algorithm 1 and the linear transformation computation in Line 8 of 1 compared to the original Gumbel-Softmax reparameterizer of the categorical distribution. Note that this additional computation complexity may not be represented as $\mathcal{O}(n)$ if we assume that there are $n$ possible outcomes by truncating the distribution, since the computation on PMF values from the inferred distribution parameter $\lambda$ depends on the PMF of distribution. Meanwhile, the implicit inference only has extra linear transformation computation in Line 5 of Algorithm 2 compared to the original Gumbel-Softmax reparameterizer of the categorical distribution, by inferring the logit of the PMF values directly in Line 3 of Algorithm 2. Hence, in this case, the additional computational complexity

Figure 8: Visualization of GENGS reparameterization of implicit inference version. Note that we do not infer the distribution parameter $\lambda$, and the PMF values are computed directly, instead.

of GENGS is $\mathcal{O}(n)$, compared to the original Gumbel-Softmax. Note that this complexity does not affect on the total complexity under the $\mathcal{O}$-notation, since the inference step of the original Gumbel-Softmax also contains the inference on the logit of the probability which has $\Omega(n)$ complexity.

---

**Algorithm 1** Explicit GENGS: Inference step on the parameter $\lambda$ of distribution $p_\lambda$

---

1: **Input:** PMF $p_\lambda(\cdot)$ of distribution D$(\lambda)$, loss function $f(\cdot)$, Gumbel-Softmax trick GS, temperature $\tau > 0$, truncation range $R$, linear transformation $\mathcal{T}(\cdot)$, constant outcome vector $c$.
2: Infer distribution parameter $\hat{\lambda}$.
3: **for** $k = 0$ **to** $n - 2$ **do**
4:     Compute $\pi_k = p_\lambda(c_k)$.
5: **end for**
6: Compute $\pi_{n-1} = 1 - \sum_{k=0}^{n-2} \pi_k$.
7: Sample $n$-dimensional one-hot-like $w \sim \text{GS}(\pi, \tau)$.
8: Compute transformation $z = \mathcal{T}(w) = \sum_j w_j c_j$.
9: Compute loss $f(z)$.
10: Update $\hat{\lambda}$ via stochastic gradient method.

---

**Algorithm 2** Implicit GENGS: Inference step on the PMF values $\pi$ of distribution $p_\lambda$

---

1: **Input:** PMF $p_\lambda(\cdot)$ of distribution D$(\lambda)$, loss function $f(\cdot)$, Gumbel-Softmax trick GS, temperature $\tau > 0$, truncation range $R$, linear transformation $\mathcal{T}(\cdot)$, constant outcome vector $c$.
2: Infer PMF logit-value $\hat{\nu}$.
3: Compute $\pi = \texttt{softmax}(\nu)$.
4: Sample $n$-dimensional one-hot-like $w \sim \text{GS}(\pi, \tau)$.
5: Compute transformation $z = \mathcal{T}(w) = \sum_j w_j c_j$.
6: Compute loss $f(z)$.
7: Update $\hat{\nu}$ via stochastic gradient method.

---

## G  DISTRIBUTIONS IN TENSORFLOW PROBABILITY

We provide the table of discrete distributions where GENGS can be applied in Table 3. We list up the discrete distributions which are available in TensorFlow Probability 0.8.0[6] in lexicographical order. Note that `Bernoulli` and `Categorical` are different from `OneHotCategorical` and `RelaxedOneHotCategorical`. The original Gumbel-Softmax are available only for

---

[6]https://www.tensorflow.org/probability/api_docs/python/tfp/distributions

(1) `OneHotCategorical` ,which is relaxed as `RelaxedOneHotCategorical`; and (2) `Bernoulli`, which is relaxed as `RelaxedBernoulli`, but can not be applied to `Categorical`. However, by generalizing the Gumbel-Softmax, i.e., GENGS, other distributions as listed below, including `Categorical`. Also, `Empirical`, which is a user-defined distribution in TensorFlow Probability, can utilize GENGS which is not in the list. Note that for `Binomial`, if $n$ is large or $p$ has extreme value close to 0 or 1, one can truncate left, right, or both. Also, for `Multinomial`, as an extension of the `Binomial` case, one can disregard unlikely samples.

Table 3: A list of distributions which can be reparameterized by GENGS with their distribution parameters in TensorFlow Probability 0.8.0.

| Distribution | PMF $P(X = k)$ | Reparameterized Parameter | Infinite Support | Truncation Side |
|---|---|---|---|---|
| `Bernoulli`$(p)$ 
 $p \in [0,1], k \in \{0,1\}$ | $p^k(1-p)^{1-k}$ | $p$ | | None |
| `Binomial`$(n,p)$ 
 $n \in \mathbb{N}_{\geq 0}, p \in [0,1], k \in \{1, \cdots, n\}$ | $\binom{n}{k}p^k(1-p)^k$ | $p$ | | None* |
| `Categorical`$(p = [p_1, \cdots, p_K])$ 
 $p_i \in [0,1], \sum p_i = 1, k \in \{1, \cdots, K\},$ 
 $k_i \in \{0,1\}, \sum k_i = 1$ | $\prod_{j=1}^K p_j^{k_j}$ | $p$ | | None |
| `DirichletMultinomial`$(n, \alpha = [\alpha_1, \cdots, \alpha_K])$ 
 $n \in \mathbb{N}, \alpha_i > 0,$ 
 $k_i \in \{0, \cdots, n\}, \sum k_i = n$ | $\frac{n! \Gamma(\sum \alpha_j)}{\Gamma(n + \sum \alpha_j)} \prod_{j=1}^K \frac{\Gamma(k_j + \alpha_j)}{k_j! \Gamma(\alpha_j)}$ | $\alpha$ | | None |
| `Geometric`$(p)$ 
 $p \in (0,1), k \in \mathbb{N}_{\geq 0}$ | $(1-p)^k p$ | $p$ | ✓ | Right |
| `Multinomial`$(n, p = [p_1, \cdots, p_K])$ 
 $n \in \mathbb{N}, p_i \in [0,1], \sum p_i = 1,$ 
 $k_i \in \{0, \cdots, n\}, \sum k_i = n$ | $\binom{n}{k_1 \cdots k_K} \prod_{j=1}^K p_j^{k_j}$ | $p$ | | None* |
| `NegativeBinomial`$(n,p)$ 
 $n \in \mathbb{N}, k \in \mathbb{N}_{\geq 0}$ | $\binom{n+k-1}{k}(1-p)^n p^k$ | $p$ | ✓ | Right |
| `OneHotCategorical`$(p = [p_1, \cdots, p_K])$ 
 $p_i \in [0,1], \sum p_i = 1,$ 
 $k_i \in \{0,1\}, \sum k_i = 1$ | $\prod_{j=1}^K p_j^{k_j}$ | $p$ | | None |
| `Poisson`$(\lambda)$ 
 $\lambda \in \mathbb{R}_{>0}, k \in \mathbb{N}_{\geq 0}$ | $\frac{\lambda^k \exp^{-\lambda}}{k!}$ | $\lambda$ | ✓ | Right, Both |
| `Zipf`$(n,s)$ 
 $s \geq 0, n \in \mathbb{N}, k \in \{1, \cdots, n\}$ | $\frac{k^{-s}}{\sum_{j=1}^n n^{-s}}$ | $s$ | | None |

## H  KL DIVERGENCE BETWEEN TWO TRUNCATAED DISTRIBUTIONS

**Theorem.** *Assume two truncated distributions $X \sim TD(\lambda, n)$ and $Y \sim TD(\hat{\lambda}, n)$ where $\pi_k = P(X = k)$, $\hat{\pi}_k = P(Y = k)$. Then, the KL divergence between $X$ and $Y$ can be represented in the KL divergence between the categorical distributions where $KL(Y\|X) = KL(Categorical(\hat{\pi})\|Categorical(\pi))$.*

*Proof.*

$$\mathrm{KL}(Y||X) = \sum_k P(Y = k) \log \left( \frac{P(Y = k)}{P(X = k)} \right) \tag{23}$$

$$= \sum_k \hat{\pi}_k \log \left( \frac{\hat{\pi}_k}{\pi_k} \right) \tag{24}$$

$$= \mathrm{KL}(\mathrm{Categorical}(\hat{\pi})||\mathrm{Categorical}(\pi)) \tag{25}$$

$\square$

## I   EXPERIMENT: GENERAL SETTING

For all experiments, we use Intel Core i7-6700K CPU, 32GB RAM, and Titan X. For the dependency, we use TensorFlow version 1.15.0, TensorFlow Probability version 0.8.0, and PyTorch version 1.0.1. Also, we run the experiments over 10 times for each experiment.

## J   SYNTHETIC EXAMPLE

### J.1   EXPERIMENTAL SETTING

In this experiment, we first sample $t_1, \cdots, t_k$ i.i.d. from a discete distribution $\mathrm{D}(\theta)$ for a fixed $\theta > 0$, and optimize the loss function $\mathbb{E}_{z \sim p(z|\lambda)} \left[ \sum_{i=1}^{k} (z_i - t_i)^2 \right]$ with respect to $\lambda$ where $p(z|\lambda)$ is $\mathrm{D}(\lambda)$. We use Poisson(20), Binomial(20, .3), Multinomial(3, [.7, .2, .1]), and NegativeBinomial(3, .4) in this experiment, and the distribution parameter which we want to infer in each distribution is $\lambda$ in Poisson($\lambda$), Binomial(20, $\lambda$), Multinomial(3, $\lambda$), and NegativeBinomial(3, $\lambda$). For GENGS, we use truncation level $(7, 36)$ and 12 for the Poisson and the negative binomial, respectively. Note that the binomial case does not require truncation of the distribution. We use $k = 5$ sampled targets for the Poisson and the binomial cases, and $k = 1$ for the negative binomial case. In this experiment, we separately utilize the temperature $\tau$ as $\tau = 1.$ for the high-temperature case, and $\tau = .25$ for the low-temperature case. To compute the variance of gradients, we sampled 100 gradients for the Poisson and binomial, and 500 gradients for the negative binomial. For fair comparisons, we use $m = 1$ fixed category for gradients for the RBs. Whereas it is able to use more than one fixed gradient in the synthetic example, if there is more than one latent dimension, $K$, it requires to compute $m^K$ gradient combinations, which has high complexity. We also adapt the Rao-Blackwellization idea in GENGS, which is utilizing $m = 1$ fixed gradient and utilizing GENGS for the remainings, namely GENGS-RB. We exclude UNORD snice UNORD fails to converge to the optimal parameter because of its approximation accuracy problem with single gradient sample.

**Closed-form True Gradient Derivation for the Poisson Synthetic Example.** Throughout the synthetic example, we compare the quality of gradient estimators by the convergence of losses, variances of estimated gradients, and biases between true gradient and estimated gradient. To compute the bias between the true gradient and the estimated gradient, we need the closed-form solution of the true gradient. We find that the Poisson case has the closed-form true gradient, and the derivation is as follows.

**Proposition.** *If $p(z|\lambda)$ is a Poisson distribution with a rate parameter $\lambda$, the true gradient of $\mathcal{L} = \mathbb{E}_{z \sim p(z|\lambda)}[(z - t)^2]$ with respect to $\lambda$ has a closed-form solution, $\frac{\partial \mathcal{L}}{\partial \lambda} = 2\lambda - 2t + 1$.*

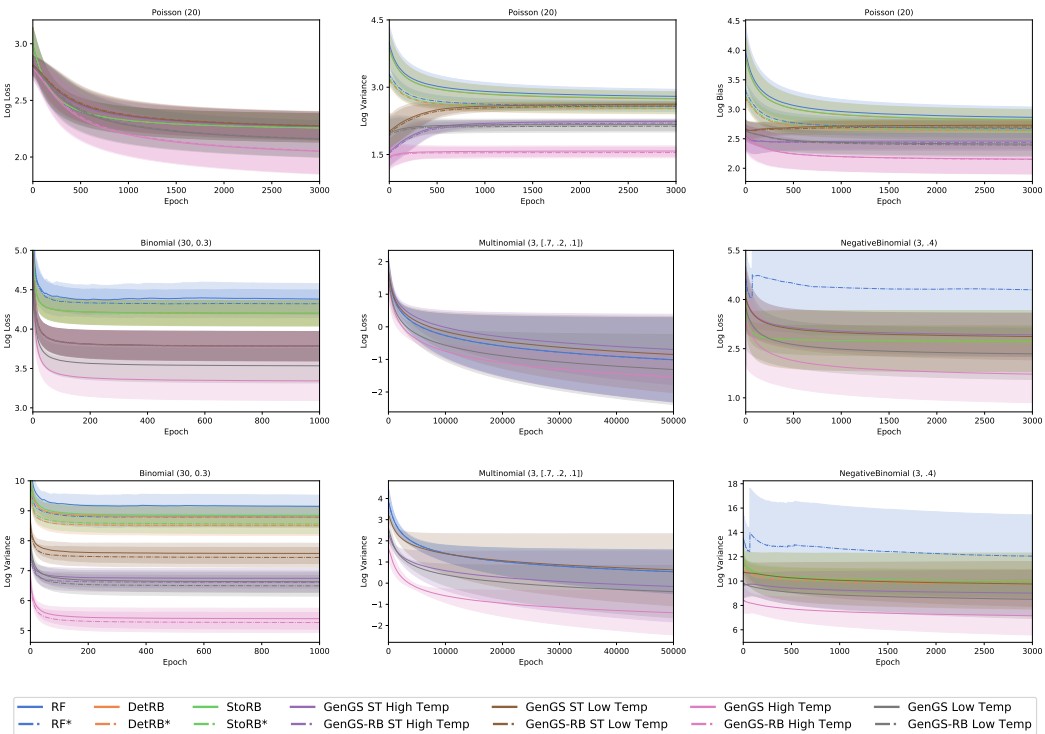

Figure 9: Synthetic example performance curves in log scale: (Top Row) Losses, variances, and biases of gradients for Poisson; (Middle Row) Losses for Binomial, Multinomial, and NegativeBinomial; (Bottom Row) Variances of gradients for Binomial, Multinomial, and NegativeBinomial. We utilize the cumulative average for smoothing the curves, and we also provide confidence intervals together.

*Proof.* Note that Poisson distribution with the rate parameter $\lambda$ has a mean $\lambda$ and a variance $\lambda$. Hence, the Poisson distribution has the first moment $\mu_1 = \lambda$, and the second moment $\mu_2 = \lambda^2 + \lambda$.

$$\frac{\partial \mathcal{L}}{\partial \lambda} = \frac{\partial}{\partial \lambda} \mathbb{E}_{z \sim p(z|\lambda)}[(z-t)^2] \tag{26}$$

$$= \frac{\partial}{\partial \lambda} \sum_{z \geq 0} p(z|\lambda)(z-t)^2 \tag{27}$$

$$= \sum_{z \geq 0} \frac{\partial}{\partial \lambda} \Big[ p(z|\lambda)(z-t)^2 \Big] \tag{28}$$

$$= \sum_{z \geq 0} \frac{\partial}{\partial \lambda} \Big[ \frac{\lambda^z e^{-\lambda}}{z!}(z-t)^2 \Big] \tag{29}$$

$$= \sum_{z \geq 0} (z-t)^2 \frac{\partial}{\partial \lambda} \Big[ \frac{\lambda^z e^{-\lambda}}{z!} \Big] \tag{30}$$

$$= t^2 \frac{\partial}{\partial \lambda} e^{-\lambda} + \sum_{z \geq 1}(z^2 - 2tz + t^2)\Big( \frac{z\lambda^{z-1}e^{-\lambda} - \lambda^z e^{-\lambda}}{z!} \Big) \tag{31}$$

$$= -t^2 e^{-\lambda} + \sum_{z \geq 0} \big((z+1)^2 - 2t(z+1) + t^2\big)\Big( \frac{\lambda^z e^{-\lambda}}{z!} \Big) - \sum_{z \geq 1}(z^2 - 2tz + t^2)\Big( \frac{\lambda^z e^{-\lambda}}{z!} \Big) \tag{32}$$

$$= -t^2 e^{-\lambda} + \sum_{z \geq 0} \big(z^2 - 2(t-1)z + (t-1)^2\big)p(z|\lambda) - \sum_{z \geq 1}(z^2 - 2tz + t^2)p(z|\lambda) \tag{33}$$

$$= -t^2 e^{-\lambda} + \left( \mu_2 - 2(t-1)\mu_1 + (t-1)^2 \right) - \left( \mu_2 - 2t\mu_1 + t^2(1 - e^{-\lambda}) \right) \tag{34}$$

$$= 2\lambda - 2t + 1 \tag{35}$$

$$\square$$

## J.2 EXPERIMENTAL RESULT

We compare the log-loss and the log-variance of estimated gradients from various estimators in this experiment. We also compare the log-bias in the Poisson case. We additionally provide Figure 9 to report the confidence interval, and Figure 10 to show the convergence of loss which may not be seen in Figure 3 of the main paper.

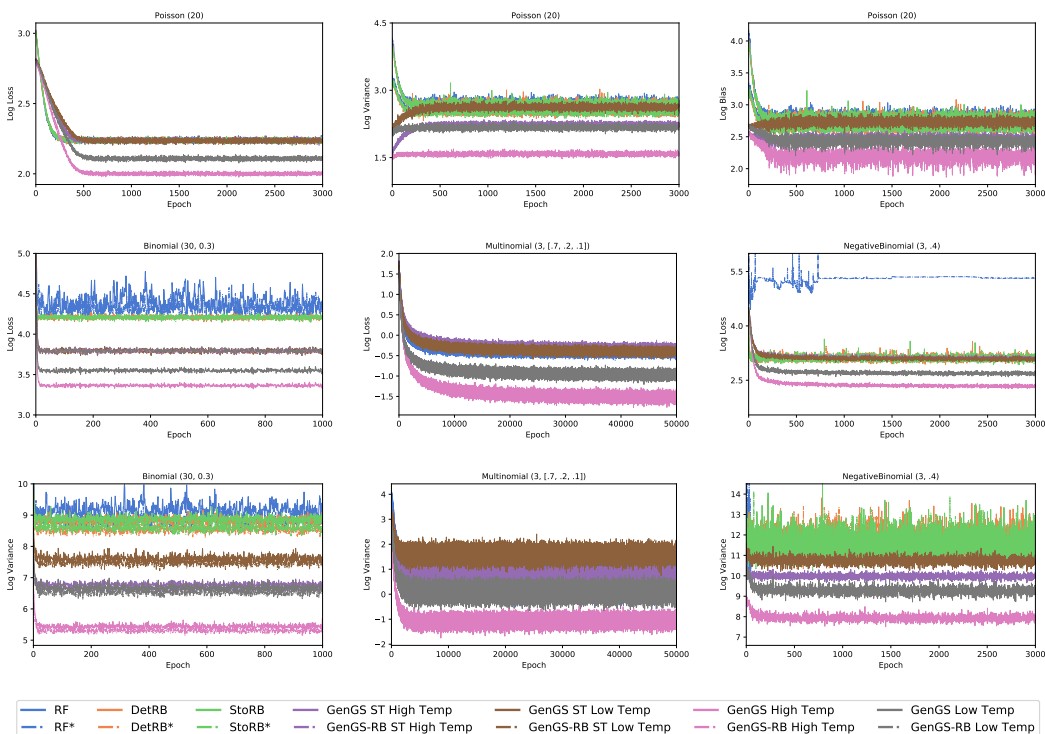

Figure 10: Synthetic example performance curves in log scale: (Top Row) Losses, variances, and biases of gradients for Poisson; (Middle Row) Losses for Binomial, Multinomial, and NegativeBinomial; (Bottom Row) Variances of gradients for Binomial, Multinomial, and NegativeBinomial. We do not smoothen the curves to show the convergence of losses in this figure.

## K   VAE: SYNTHETIC EXPERIMENT ON DEEP GENERATIVE MODELS

### K.1 EXPERIMENTAL SETTING

We utilize the (truncated) Poisson, the (truncated) geometric, and the (truncated) negative binomial distributions in this experiment. Both MNIST and OMNIGLOT[7] are hand-written gray-scale datasets of size $28 \times 28$. We split MNIST dataset into $\{\text{train} : \text{validation} : \text{test}\} = \{45,000 : 5,000 : 10,000\}$, and OMNIGLOT dataset into $\{22,095 : 2,250 : 8,070\}$.

We construct two fully-connected hidden layers of dimension 500 for the encoder and the decoder, and we set the latent dimension $K = 50$ for both MNIST and OMNIGLOT datasets. we use $\texttt{tanh}$ activation function, learning rate $\texttt{5e-4}$, training epoch 500, and batch size 100, 45 for MNIST and

---

[7]https://github.com/yburda/iwae/tree/master/datasets/OMNIGLOT

OMNIGLOT, repectively. For GENGS, we use exponential temperature annealing[8] from 1. to .5, and truncation levels (1) 12 for Poisson(2); (2) 15 for Poisson(3); (3) 25 for Geometric(.25); (4) 15 for Geometric(.5); (5) 30 for NegativeBinomial(3, .5); and (6) 30 for NegativeBinomial(5, .3).

## K.2 EXPERIMENTAL RESULT

Table 4 shows the negative ELBO results on the VAE experiments with a full range of gradient estimators. The variants of GENGS show the lowest negative ELBO in general. We empirically found that the extreme probability imbalance, due to the explicit PMF restriction, induces unstable learning which leads to the performance degradation of RELAX or REBAR. We found that utilizing Straight-Through for the discretization degrades the performance. Meanwhile, the implicit inference methodology further leads to better optimal points, enabled by loosening the PMF condition. The empirical reason why the implicit version is better than the explicit version is that the inferred PMF shape is thinner in the implicit case.. Hence, the implicit distribution has lower variance than the explicit one, and consequently samples consistent values that lead to better trained neural network parameters, although it looses the original PMF shape.

Table 4: Test negative ELBO on MNIST and OMNIGLOT datasets. The lower is better for the negative ELBO. Symbol "—" indicates no convergence.

| MNIST | RF | RF* | NVIL | MUPROP | VIMCO(5) | REBAR | RELAX |
|---|---|---|---|---|---|---|---|
| Pois(2) | $153.36_{\pm3.23}$ | $122.81_{\pm2.41}$ | $129.34_{\pm4.72}$ | $125.43_{\pm2.27}$ | $122.55_{\pm3.28}$ | $123.44_{\pm2.54}$ | $122.71_{\pm1.92}$ |
| Pois(3) | $153.34_{\pm4.22}$ | $123.12_{\pm2.21}$ | $130.24_{\pm3.32}$ | $125.92_{\pm1.81}$ | $121.15_{\pm2.57}$ | $120.62_{\pm2.31}$ | $119.84_{\pm2.18}$ |
| Geom(.25) | $168.86_{\pm3.15}$ | $127.90_{\pm1.97}$ | $135.90_{\pm2.38}$ | $137.90_{\pm2.14}$ | $127.21_{\pm2.55}$ | $135.12_{\pm2.74}$ | $136.80_{\pm3.06}$ |
| Geom(.5) | $165.00_{\pm2.93}$ | $129.20_{\pm2.03}$ | $138.47_{\pm2.30}$ | $136.40_{\pm1.78}$ | $129.91_{\pm2.90}$ | $138.37_{\pm2.98}$ | $139.41_{\pm3.59}$ |
| NegBin(3,.5) | — | $116.67_{\pm5.97}$ | $119.28_{\pm7.80}$ | $131.96_{\pm6.49}$ | $112.69_{\pm4.30}$ | — | — |
| NegBin(5,.3) | — | $130.03_{\pm3.99}$ | $133.44_{\pm4.27}$ | $144.05_{\pm8.15}$ | $124.48_{\pm2.72}$ | — | — |

| MNIST | STORB* | UNORD | GENGS ST (Ex.) | GENGS (Ex.) | GENGS ST (Im.) | GENGS (Im.) |
|---|---|---|---|---|---|---|
| Pois(2) | $124.02_{\pm4.91}$ | $153.94_{\pm5.01}$ | $130.48_{\pm1.51}$ | $103.18_{\pm0.92}$ | $106.92_{\pm2.60}$ | $96.04_{\pm1.44}$ |
| Pois(3) | $124.41_{\pm5.96}$ | $153.65_{\pm4.83}$ | $129.86_{\pm1.77}$ | $105.15_{\pm1.71}$ | $106.66_{\pm2.46}$ | $96.01_{\pm1.27}$ |
| Geom(.25) | $131.09_{\pm4.95}$ | — | $120.46_{\pm1.47}$ | $98.43_{\pm0.81}$ | $108.14_{\pm2.67}$ | $92.52_{\pm1.62}$ |
| Geom(.5) | $139.67_{\pm2.42}$ | — | $119.10_{\pm1.09}$ | $100.92_{\pm1.24}$ | $106.44_{\pm2.78}$ | $93.81_{\pm1.60}$ |
| NegBin(3,.5) | $114.36_{\pm4.12}$ | — | $117.23_{\pm1.54}$ | $98.58_{\pm1.27}$ | $102.44_{\pm3.75}$ | $94.52_{\pm1.52}$ |
| NegBin(5,.3) | $128.02_{\pm2.60}$ | — | $122.61_{\pm2.31}$ | $100.88_{\pm2.35}$ | $102.70_{\pm3.64}$ | $95.37_{\pm1.43}$ |

| OMNIGLOT | RF | RF* | NVIL | MUPROP | VIMCO(5) | REBAR | RELAX |
|---|---|---|---|---|---|---|---|
| Pois(2) | $165.77_{\pm3.67}$ | $139.47_{\pm3.29}$ | $148.01_{\pm4.19}$ | $142.95_{\pm1.32}$ | $138.73_{\pm3.42}$ | $138.12_{\pm3.26}$ | $137.56_{\pm2.94}$ |
| Pois(3) | $164.86_{\pm4.19}$ | $140.54_{\pm2.36}$ | $148.13_{\pm3.98}$ | $143.85_{\pm1.54}$ | $139.37_{\pm3.10}$ | $137.92_{\pm3.07}$ | $137.42_{\pm2.96}$ |
| Geom(.25) | $171.79_{\pm4.59}$ | $142.68_{\pm2.96}$ | $153.69_{\pm2.52}$ | $152.17_{\pm1.77}$ | $142.94_{\pm3.96}$ | $146.78_{\pm3.62}$ | $148.91_{\pm4.03}$ |
| Geom(.5) | $170.96_{\pm3.10}$ | $142.70_{\pm1.77}$ | $153.20_{\pm1.49}$ | $149.76_{\pm2.19}$ | $142.05_{\pm3.56}$ | $149.63_{\pm3.49}$ | $151.97_{\pm3.90}$ |
| NegBin(3,.5) | — | $141.44_{\pm2.20}$ | $144.44_{\pm2.78}$ | $147.78_{\pm4.49}$ | $141.89_{\pm3.84}$ | — | — |
| NegBin(5,.3) | — | $144.44_{\pm3.68}$ | $159.40_{\pm5.13}$ | $152.81_{\pm3.34}$ | $150.49_{\pm4.09}$ | — | — |

| OMNIGLOT | STORB* | UNORD | GENGS ST (Ex.) | GENGS (Ex.) | GENGS ST (Im.) | GENGS (Im.) |
|---|---|---|---|---|---|---|
| Pois(2) | $139.61_{\pm5.87}$ | $166.64_{\pm5.13}$ | $148.60_{\pm1.98}$ | $127.89_{\pm1.44}$ | $134.76_{\pm2.65}$ | $118.17_{\pm2.22}$ |
| Pois(3) | $140.05_{\pm3.68}$ | $166.73_{\pm6.71}$ | $147.79_{\pm1.21}$ | $131.53_{\pm1.76}$ | $135.14_{\pm2.25}$ | $119.15_{\pm1.92}$ |
| Geom(.25) | $143.10_{\pm3.91}$ | — | $146.54_{\pm1.18}$ | $115.23_{\pm2.00}$ | $135.61_{\pm3.22}$ | $107.79_{\pm2.84}$ |
| Geom(.5) | $142.56_{\pm2.97}$ | — | $141.23_{\pm1.22}$ | $115.14_{\pm2.43}$ | $136.02_{\pm3.63}$ | $108.48_{\pm2.78}$ |
| NegBin(3,.5) | $129.48_{\pm4.34}$ | — | $142.33_{\pm1.92}$ | $118.57_{\pm2.71}$ | $135.02_{\pm3.99}$ | $117.02_{\pm2.18}$ |
| NegBin(5,.3) | $151.30_{\pm3.98}$ | — | $145.27_{\pm2.18}$ | $119.57_{\pm2.02}$ | $134.54_{\pm3.27}$ | $117.54_{\pm2.76}$ |

# L  TOPIC MODEL APPLICATION

## L.1  EXPERIMENTAL SETTING

*Deep Exponential Families* (DEFs) (Ranganath et al., 2015) are probabilistic graphical model which utilize the stacks of exponentail family distributions. If we assume the Poisson distribution, which is included in the exponential family, each $k^{\text{th}}$ Poisson latent variable counts the number of sub-topics occurrence.

The relationship between the super-topic and the sub-topic is modeled with the linked weights, which has positive values. Hence, with Poisson DEF, we can model hierarchical relations between

---

[8]For GENGS ST, the temperature annealing is unnecessary as the ST Gumbel-Softmax estimator does (Jang et al., 2017).

super-topics and sub-topics including the vocabularies. Here, we utilize the idea of Miao et al. (2016; 2017); Srivastava & Sutton (2017), the neural variational architecture, to extract the latent document representation as (relaxed) counts of supermost-topic, and consequently capture sub-topic counts. To ensure the positive linked weights between super-topics and sub-topics, we utilize absolute value function.

The generative process of NVPDEF is

$$z_1 \sim \text{Poisson}(\lambda_0), \ z_2 \sim \text{Poisson}(\lambda_1), \ \cdots, \ z_K \sim \text{Poisson}(\lambda_{K-1}), \tag{36}$$

$$x \sim \text{MultinomialLogisticRegression}(\lambda_K) \tag{37}$$

where we adopt multinomial logistic regression from NVDM (Miao et al., 2016), and the inference process of NVPDEF is

$$\hat{\lambda}_0 = \text{MLP}(x), \ \hat{\lambda}_1 = W_1\hat{\lambda}_1, \ \cdots, \ \hat{\lambda}_K = W_{K-1}\hat{\lambda}_{K-1} \tag{38}$$

so that the approximate Poisson posterior $q(z_k|z_{k-1})$ has $\hat{\lambda}_k$ as distribution parameter. Here, each $z_k \sim \text{Poisson}(\lambda_{k-1})$ represents the count distribution of topics from the super-topic. Each component of $W_k$, $w_{k,i,j}$ is positive, and $w_{k,i,j}$ captures the positive weight of relationship between super-topic $i$ of the $k^{\text{th}}$ layer and sub-topic $j$ of the $(k+1)^{\text{th}}$ layer. The objective function, ELBO, of NVPDEF is

$$\mathcal{L} = \mathbb{E}_{q(z_K,\cdots,z_1)}[\log p(x|z_K,\cdots,z_1)] - \sum_{k=1}^{K} \text{KL}(q(z_k|z_{k-1})||p(z_k)) \tag{39}$$

where $z_0 = x$ for simplifying the equation.

20Newsgroups[9] and RCV1-V2[10] datasets are used in this experiment. 20Newsgroups dataset has $\{\text{train} : \text{test}\} = \{11,258 : 7,487\}$ split with the vocabulary size of $2,000$, and RCV1-V2 has $\{\text{train} : \text{test}\} = \{794,414 : 10,000\}$ split with the vocabulary size of $10,000$. For the data pre-processing, stopwords are removed and the most frequent vocabularies are chosen. Especially for 20Newsgroups, we use the vocabulary from Srivastava & Sutton (2017).

For the single-stacked version of NVPDEF, we do not anneal the temperature, instead, we set temerature $\tau = .5$. For the multi-stacked version of NVPDEF, i.e., MULTI-STACKED NVPDEF, we utilized 10-sample on the latent layers for the stable optimization of consecutive sampling. Also, to have better chances of learning, we utilize linear temperature annealing from $\tau = 3.$ to $\tau = .5$ during the training period. For all neural network models, we utilize two 500-dimensional hidden layers for the encoders. We use $50, 20\text{-}50$ stacked layers for 20Newsgroups, $200, 50\text{-}200$ stacked layers for RCV1-V2 dataset. We set $\lambda_1 = .75$ with truncation level 15 for the single-stacked case, and $\lambda_1 = 1.1, \lambda_2 = 1.$ with truncation level 15 for the multi-stacked case. We train NVPDEF for 100 epochs with batch size 256 and learning rate `1e-3`. We also iteratively update encoder parameters and each linked weight parameter of latent variable. As a performance measure, we utilize perplexity $\texttt{perp} = \exp(-\frac{1}{D}\sum_d \frac{\log p(d)}{N_d})$ where $N_d$ is the number of words in document $d$, and $D$ is the total number of documents.

### L.2 EXPERIMENTAL RESULT

We provide super-topic, sub-topic, and word relationship obtained from two-layer-stacked NVPDEF in 20Newsgroups dataset by listing up the top-weighted sub-topics and words in Table 5.

## M  OPEN RESEARCH QUESTION: NON-PARAMETRIC REPARAMETERIZATION TRICK

In GENGS, to reparameterize discrete distributions, we convert *sampling process* to *categorical selection process* by finitizing the support of the distribution with truncation. Here, truncating distribution converts *categorical selection on countably finite number of categories* to *categorical selection on finite number of categories*, i.e., turn non-parametric problem into parametric problem.

---

[9] http://qwone.com/~jason/20Newsgroups/
[10] https://trec.nist.gov/data/reuters/reuters.html

Table 5: Super-topic, sub-topic, and word relationship obtained from two-layer-stacked NVPDEF in 20Newsgroups dataset.

| Super Topic 1 | | Super Topic 2 | | Super Topic 3 | | Super Topic 4 | | Super Topic 5 | | Super Topic 6 | |
|---|---|---|---|---|---|---|---|---|---|---|---|
| Topic 1 | Topic 2 | Topic 3 | Topic 4 | Topic 5 | Topic 6 | Topic 7 | Topic 8 | Topic 9 | Topic 10 | Topic 11 | Topic 12 |
| lebanese | hitler | knife | tire | water | probe | brand | honda | flyers | pitcher | holy | bible |
| lebanon | jewish | gun | helmet | air | spacecraft | outlet | dealer | braves | montreal | resurrection | biblical |
| palestinian | nazi | police | rider | heat | plane | sale | offer | hitter | score | passage | faith |
| arabs | religion | weapon | bike | cold | shuttle | shipping | sell | philadelphia | season | jesus | prayer |
| israeli | territory | firearm | gear | oil | nasa | insurance | condition | detroit | game | worship | doctrine |
| islamic | sentence | handgun | motorcycle | gas | launch | price | purchase | rangers | player | christ | verse |
| regulation | moral | officer | wheel | noise | fuel | supply | market | minnesota | tie | sin | god |

The categorical selection on finite number of categories by disregarding the samples of extremely small probability might cause a problem if we need to utilize full range of possible outcomes. For example, in the multinomial case in Appendix E.4, as $n$ and $k$ grows, proposed GENGS should ignore numerous probable samples due to the high complexity on the number of possible outcomes.

A* sampling (Maddison et al., 2014) is a non-parametric version of Gumbel-Max trick which we also utilize in GENGS, since A* sampling searches maximum Gumbel sample among countably infinite Gumbel samples by A* algorithm. Utilizing A* sampling concept in reparameterizing the distribution with countably infinite support could lead to better reparameterization in terms of reparameterizing the exact distribution instead of approximate distribution. However, we utilize the truncated distribution in proposed GENGS to convert countably infinite categorical selection into finite categorical selection for the following reasons.

First, while adapting non-parametric methodology into a neural network, which has a fixed number of parameters, people usually give a limit as a certain point by utilizing the truncation level. An example of such a case is a Stick-breaking VAE (Nalisnick & Smyth, 2017) which utilizes Dirichlet process in the latent variable of VAE, and the authors finitized the number of sticks by the human guidance. Second, while there is no previous work on reparameterization trick for fully-non-parametric categorical selection, if we finitize the number of categories with the truncated distributions suggested in the paper, we can utilize Gumbel-Softmax reparameterizer which already verified in deep generative model community and widely used by the implementation in the deep learning framework such as TensorFlow, i.e., `RelaxedOneHotCategorical`. Finally, if we have to choose one between the parametric model and the non-parametric model, the choice depends on the situation that we face up to. For example, we can compare Gaussian mixture model (GMM) and Dirichlet process Gaussian mixture model (DPGMM). If we have a clue on the number of clusters, we could directly apply GMM instead of DPGMM. However, if we know nothing about the data, utilizing DPGMM can be a good choice. Also, we can not directly compare GMM and DPGMM along the same line, since the experimental result differs from data to data.

In summary, as other non-parametric models do, we turn the non-parametric problem into the parametric problem, especially by utilizing the truncated distribution, and this kind of treatment is a natural way of solving such difficulty. However, we believe that investigating non-parametric reparameterizer, particularly utilizing A* sampling which is theoretically solid, is a crucial and open research question in the deep generative model community.

