# OpenReview forum: "Generalized Gumbel-Softmax Gradient Estimator for Generic Discrete Random Variables"
_ICLR.cc/2021/Conference — Reject_

### Official Review · AnonReviewer2 · 2020-10-26
**Marginal contribution, unclear what benefits are**

**Rating:** 4
**Confidence:** 5

**Review:**

This paper presents a new methodology for inference of discrete variables in computational graphs.

Overall the paper represents a substantial amount of work, and it is clearly written. Nonetheless, I believe it is not significant enough so I can recommend acceptance.

1)Theoretical analysis, while pertinent, is mostly elementary and unsurprising. I would have expected something closer to rates of convergence but the kind of analysis presented (almost sure convergence when the truncation size goes to infinity) is essentially  obvious for anyone who was taken an elementary course on probability/measure theory. I would expect theoretical results to enlighten some aspects of their methods, but that is absent.

2)I am in general unconvinced by the method. I don't think there is anything specially novel besides truncation. But just proposing truncation is not enough merit, I believe, as truncation is a natural choice. I would have expected an analysis of how truncation affects the results.

3)I am not convinced by experimental results. Authors don't include Gumbel-Softmax as a baseline and I wonder whether any goodness in reported results (Besides the topic model) is a consequence of the action of Gumbel-Softmax instead of their method.

4)The topic modeling experiment is interesting, since it involves a poisson distribution, with infinitely many possible values. However,  there, the relevant comparison is missing. This alternative could be, for example, the detr method (Liu et al, 2019). Notice that I disagree with the authors claim that this method is doing something completely unique. The Liu et al paper can be used with similar purposes.

---

### Official Review · AnonReviewer4 · 2020-10-27
**Interesting approach, but have concerns about significance**

**Rating:** 4
**Confidence:** 3

**Review:**

Summary: The paper presents a generalization of the Gumbel-Softmax gradient estimator. The original Gumbel-Softmax is usually applied to Bernoulli and categorical random variables. The method proposed in the paper attempts to extend it applicability to other discrete distributions, such as Poisson, multinomial, geometric, among others. The main ideas of the approach are: (1) Random variables that may take countably infinite values are truncated, (2) The sampling process of the random variable is converted to a one-hot scheme (where Gumbel-Softmax relaxation is applied), (3) ``One-hot'' samples are reverted to the original sample space.

Clarity: While there's some room for improvement, I think the paper is clear.

Pros:
- The paper tackles a very relevant problem; namely, estimating gradients with respect to the parameters of general discrete random variables. These distributions are used in many scenarios, and some of the existing methods are not always applicable. This paper aims to extend some of the current methods (specifically, Gumbel-Softmax) to general discrete distributions.

Cons:
I have concerns about the method's significance.
- The main claim is that the proposed method generalizes the Gumbel-Softmax estimator to general discrete distributions. However, I don't see how the method leads to an efficient algorithm for distributions with some combinatorial component. For instance, take the multinomial distribution with parameters (n, k). The number of possible outcomes is finite, but grows fast with n and k. Without truncation, the method can be applied by assigning each possible outcome to a one-hot vector, which becomes very large for moderate values of n and k. An alternative would involve removing potential outcomes with very low probability (i.e., truncating, as suggested in the paper). However the paper does not explain how to apply the truncation in cases like this. The only experiment in the paper with multinomial distributions involves small values of n and k (3 for both), and thus this issue does not arise.
- The multinomial distribution is just one example where the issue mentioned above occurs. There are several other distributions for which this happens (e.g. a discrete distribution over tree structures), and it seems that no solution is proposed. If I understood the approach correctly, and this is the case, I think that a discussion of this should be included in the main paper. Is there a way to deal with this using the proposed approach?
- For other non-combinatorial cases (e.g. Poisson, geometric), where the proposed approach may be applied efficiently, the method reduces to truncating the support of the random variable, which may be seen as a simple heuristic.

In summary, as presented, I think the paper does not provide an efficient approach to deal with general discrete distributions, and reduces to a simple heuristic for some of the simpler cases mentioned above. I think extending the approach to other more challenging distributions (multinomial, etc) could be an interesting direction to explore.

Recommendation: Reject.

---

### Official Review · AnonReviewer3 · 2020-10-27
**Simple idea of using Gumbel-Softmax to sample from discrete distributions, experimental part can be expanded and improved**

**Rating:** 5
**Confidence:** 3

**Review:**

**summary**
the paper proposes a method of learning discrete approximate posterior distributions with potentially unbounded support. The idea is to truncate it to a finite set of states and use Gumbel-Softmax relaxation for samples from the truncated distribution.
The approach is illustrated on VAE example as well as topic modelling.

**pros**
The idea of using Gumbel-Softmax to relax samples from another discrete distribution isn't new, however the application of this idea to distribution discussed in this paper does look novel. Gumbel-Softmax estimator is known to be computationally efficient and performing well in many situations, so it's perhaps not surprising that it performs well in the current context. Experimental results in topic modelling look very impressive.

**cons**
* I found it hard to follow the experimental section. In Section 7.2 the authors reference Figurnov et al (2018) for the setup however it seems that the reference only discusses continuous distributions. It is not clear what is the form of the prior and approximating posterior used in VAE experiments. This makes it hard to evaluate the paper.
* I found the proofs that proposed approximation converges to true samples to be of little value: they do not give quantitative bounds on what the truncation region should be for given sample quality.
* I think that comparison with StoRB and UnOrd estimators using only m=1 is not enough, it is possible that increasing m, although adding more computation, will significantly improve overall performance of the final LL.

comments:
* My suggestion for improving the paper would be to provide much more experimental details, regarding the setup, datasets and baselines in the main text of the paper. At the same time the theoretical part can be reduced given that the idea is very straightforward.

---

### Official Review · AnonReviewer1 · 2020-10-30
**Simple method and empirical results support claim but writing needs work**

**Rating:** 4
**Confidence:** 3

**Review:**

Summary:
* This paper addresses the issue of applying continuous relaxations to simple countably infinite discrete distributions such as the Poisson or Geometric distributions.
* The proposed approach truncates discrete distributions and applies a continuous relaxation to the truncated subset, i.e. the Gumbel-Softmax trick.
* The experiments demonstrate the method outperforms other estimators that do not apply continuous relaxations.

Strengths:
* The method is straightforward and is a good first step towards applying continuous relaxations to high-dimensional discrete distributions.
* The approach demonstrates improvements in the test likelihood in the presented experiments.

Weaknesses:
* A contiguous range must be chosen. This has clear drawbacks if there is no obvious structure as in the simple distributions considered in this paper (Pois, Geom, NegBin).
* Too much space was dedicated to explaining the method twice, resulting in a lack of space for experiments. The MNIST/Omniglot experiments clearly show that the proposed estimator performs well by fixing the model and varying over different gradient estimators. What is the purpose of the topic modeling experiments? Are the baselines comparable to the proposed model?

Decision: Reject
* The method is simple and extends the Gumbel-Softmax trick to countably infinite discrete distributions.
* The empirical results appear to be quite good compared to the reported baselines, although I am not familiar with the datasets used or whether the baselines are fair comparisons.
* However, the writing needs a lot of work. Most of the experimental details have been relegated to the appendix because of space used to explain the method twice.

Suggestions:
* The truncation seems unnecessary. Is there a tweak to the method similar to Liu et al [1], such as applying the Gumbel-Softmax trick to the largest atoms (such that their total mass > 1 - 0.0005)? Or exploring learned partitions? Additionally, Liu et al [1] correct for bias via sampling. Is there a similar extension here? For example the (n+1)-th value of c can be sampled randomly from the complement of the support subset C.
* Sections 3 and 4 are redundant. Get rid of section 3 and, if necessary, simplify the formal description of the method so that it is understandable without section 3. Find the minimal way to convey the method while remaining comprehensive. This should give more room to fully and clearly explain the experiments.
* Given the simplicity of the method (which is a good thing), is all of the notation necessary?
* In addition to the current results on topic modeling, I would like to see the fixed NVPDEF model trained with some of the gradient estimators from the MNIST/OMNIGLOT experiments.

[1] R. Liu, J. Regier, N. Tripuraneni, M. I. Jordan, and J. McAuliffe. Rao-blackwellized stochastic gradients for discrete distributions. International Conference on Machine Learning, 2019.

---

### Decision · Program_Chairs · 2021-01-07
**Final Decision**

**Decision:**

Reject

**Comment:**

All reviewers agree the paper does not meet the acceptance bar, and an authors' rebuttal is not available. Therefore, I recommend rejection.